# Multi-omic brain and behavioral correlates of cell-free fetal DNA methylation in macaque maternal obesity models

Benjamin I. Laufer[1,2,3,12], Yu Hasegawa [4,13], Zhichao Zhang [4,13], Casey E. Hogrefe[5], Laura A. Del Rosso[5], Lori Haapanen[3], Hyeyeon Hwang [1,2,3], Melissa D. Bauman[3,5,6,7], Judy Van de Water [7,8], Ameer Y. Taha[4], Carolyn M. Slupsky [4,7,9], Mari S. Golub[5], John P. Capitanio [5,10], Catherine A. VandeVoort[5,11], Cheryl K. Walker[3,5,7,11] & Janine M. LaSalle [1,2,3,7] ✉

Maternal obesity during pregnancy is associated with neurodevelopmental disorder (NDD) risk. We utilized integrative multi-omics to examine maternal obesity effects on offspring neurodevelopment in rhesus macaques by comparison to lean controls and two interventions. Differentially methylated regions (DMRs) from longitudinal maternal blood-derived cell-free fetal DNA (cffDNA) significantly overlapped with DMRs from infant brain. The DMRs were enriched for neurodevelopmental functions, methylation-sensitive developmental transcription factor motifs, and human NDD DMRs identified from brain and placenta. Brain and cffDNA methylation levels from a large region overlapping *mir-663* correlated with maternal obesity, metabolic and immune markers, and infant behavior. A *DUX4* hippocampal co-methylation network correlated with maternal obesity, infant behavior, infant hippocampal lipidomic and metabolomic profiles, and maternal blood measurements of *DUX4* cffDNA methylation, cytokines, and metabolites. We conclude that in this model, maternal obesity was associated with changes in the infant brain and behavior, and these differences were detectable in pregnancy through integrative analyses of cffDNA methylation with immune and metabolic factors.

In North America, more than half of pregnant women are considered to be overweight or obese[1,2]. Maternal obesity and related metabolic conditions are associated with a significantly increased risk of offspring with neurodevelopmental disorders (NDD), including autism spectrum disorders (ASD)[3–7]. NDDs are increasing in prevalence[8], and ASD is currently diagnosed in 1 in 54 children in the United States of America, where the diagnosis is ~4× more prevalent in males than females[9]. The elevated risk of an NDD/ASD resulting from maternal obesity is hypothesized to be related to a complex cascade of metabolic and inflammatory events that alter developmental gene regulatory networks. In mice, maternal obesity is associated with sex-specific differences in embryonic brain gene expression, affecting genes related to immunity and inflammation, metabolism, oxidative stress, and development[10]. Also in mice, maternal high-fat diet results in differences in maternal metabolism and inflammation that alter adult offspring brain inflammation and behavior[11,12]. In Japanese macaques, maternal high-fat diet resulted in metabolic and cytokine differences with long-lasting effects on offspring behavior[13]. In humans, altered metabolites have been observed in the serum of mothers of young children with ASD, including those related to the one-carbon metabolism, critical for the epigenetic modification DNA cytosine methylation[14]. In mice, a perinatal high-fat diet was found to

*A list of author affiliations appears at the end of the paper. ✉ e-mail: jmlasalle@ucdavis.edu

alter one-carbon metabolism and DNA methylation in the prefrontal cortex of male offspring[15]. The effect of maternal obesity on one-carbon metabolism in offspring was also seen in baboons fed a high-fat, high-energy diet[16]. Together, these findings demonstrate that maternal diet alters cytokine and metabolic profiles during pregnancy and suggest that these contribute to altered behavior and brain DNA methylation profiles in offspring from obese dams. However, challenges remain in understanding the epigenetic mechanisms that explain inter-individual differences following exposure to maternal obesity in humans.

As the maternal-fetal interface, the placenta is the fetal organ that is first affected by the altered metabolites and cytokines resulting from maternal obesity. In humans, pre-pregnancy obesity associated with elevated inflammatory cytokine levels in maternal serum and differential expression of genes related to nutrient transport and immunity in the placenta[17]. Women with obesity have a higher risk of developing gestational hypertension and preeclampsia[18]. Maternal pre-pregnancy obesity and trimester-specific gestational weight gain is associated with differential DNA CpG methylation in the placenta[19]. Maternal obesity is also associated with differential CpG methylation and expression of adiponectin and leptin genes in human placenta[20]. Adiponectin and leptin are adipokines, which are cytokines secreted by adipose tissue that function as cellular signaling molecules, and their overexpression results in inflammation and altered metabolism[21,22]. Alterations to inflammatory cytokine levels, which include adipokines, have been associated with ASD[23–25], and cytokines play a critical role at the placenta[26,27]. Functionally, the placental DNA methylome retains profiles of early embryonic development, including neurodevelopment[28–30]. Previously, we have shown that genome-wide DNA methylation profiles can distinguish human placental samples from newborns later diagnosed with ASD compared to typically developing controls and that DNA methylation profiles are shared between placenta and embryonic brain in a mouse model of a human NDD/ASD relevant environmental exposure[31–33]. This epigenetic convergence between placenta and brain suggests that placental DNA methylation can inform about individual NDD/ASD risk.

Ideally, epigenetic biomarkers of individual NDD/ASD risk would be obtained during pregnancy in order to design behavioral and therapeutic strategies for improved child outcomes. However, direct fetoplacental sampling is invasive and increases the risk for pregnancy loss. For this reason, non-invasive prenatal testing (NIPT) has become an increasingly attractive option for fetal diagnostics[34]. NIPT is based on assaying the cell-free fetal DNA (cffDNA) that circulates in the blood of pregnant mothers[35,36]. Genetic evidence from cases of anembryonic pregnancies or confined placental mosaicism have demonstrated that cffDNA originates from the trophoblasts of the placenta[37–39]. Epigenetic evidence has confirmed the placental origin of cffDNA through the detection of hypomethylated domains called partially methylated domains[40,41], which are uniquely characteristic of placenta in individuals without cancer[30,42]. cffDNA is generated by developmental apoptosis, during turnover of the syncytiotrophoblast, and is released into maternal circulation as a membrane bound entity[43,44]. Furthermore, cffDNA contains a DNA methylation profile representative of its placental origin[41,45,46]. cffDNA represents between 12–41% of cell-free DNA (cfDNA) in the plasma of pregnant women with the percent contribution increasing throughout pregnancy and the remainder of the profile originating from neutrophils, lymphocytes, and the liver[40].

## Results

### Rhesus macaque maternal obesity models

The model was based on naturally obese rhesus macaque dams that we previously demonstrated produce offspring with a relevant neurobehavioral profile[47]. In addition to comparing obese ($n = 7$) to lean matched controls ($n = 6$), we also examined the effects of dietary (caloric

restriction, $n = 5$) and pharmacological (pravastatin, $n = 7$) obesity interventions, and examined a total of 25 male offspring. Pravastatin was chosen as it is clinically used to lower blood pressure, and has received recent attention as it may attenuate gestational hypertension and the risk of developing preeclampsia[48]. Compared to obese control and both obese treatment groups, the lean control group showed a significantly ($p < 0.05$) reduced pre-pregnancy body composition scores (BCS) and significantly reduced body weights at all four gestational timepoints (Supplementary Table 1).

In order to characterize the multi-factorial molecular cascade that results from maternal obesity at the individual level, we generated longitudinal cffDNA methylomes from four pregnancy timepoints across all trimesters as well as three infant brain regions (hippocampus, prefrontal cortex, and hypothalamus) at 6 months old (Fig. 1a). We integrated the DNA methylome results obtained from whole-genome bisulfite sequencing (WGBS) with immunological and metabolomic assays of maternal blood across pregnancy (Supplementary Data 1), two behavioral tests assessing infant social and abstract stimuli recognition memory that is relevant to an NDD/ASD, and infant brain lipidomics and metabolomics (Supplementary Data 2).

### cffDNA methylation profiles are consistent with the placental methylome

First, to assess the quality of the cffDNA methylomes, we performed three analyses to confirm that we could recapitulate previous findings. To confirm that the DNA methylation profile of the cffDNA represents its placental origin more closely than cfDNA in a non-pregnant female, we performed a pilot experiment to compare the DNA methylation profiles of the three sample sources (placental biopsies, cffDNA, and cfDNA) in lean macaques. Principal component analysis (PCA) of the average smoothed methylation levels from regulatory regions and gene bodies revealed that the cffDNA shows a closer relationship to the placenta than cfDNA (Supplementary Fig. 1a). Next, using the primary samples for the study, we leveraged the fact that the pregnancies were all screened to be male fetuses and utilized the ratio of reads from the Y and X chromosomes to show that the fetal fraction of the cffDNA increased throughout the different trimesters of pregnancy (Supplementary Fig. 1b). Lastly, using a 20 kb window approach to assess global methylation distributions, we compared cffDNA, cfDNA, and brain WGBS datasets from the main experiment, and showed that cffDNA was hypomethylated compared to both and cfDNA and brain (Supplementary Fig. 1c). Taken together, these results demonstrate that the cffDNA methylation profiles are consistent with the fetal origin of the placental methylome[42].

### Both cffDNA and brain DMRs map to genes involved in neurodevelopment, cellular adhesion, and cellular signaling

In order to test the hypothesis that maternal obesity and interventions alter DNA methylation patterns in cffDNA and brain, we performed pairwise contrasts of obese vs. control, (caloric) restriction vs. obese, and pravastatin vs. obese, for the cffDNA samples from maternal blood during trimester 1 (GD45), trimester 2 (GD90), early trimester 3 (GD120), and late trimester 3 (GD150) as well as brain region samples from the hippocampus, prefrontal cortex, and hypothalamus of the same infants at 6 months. Each pairwise DMR comparison (empirical $p < 0.05$) generated background regions with similar gene length and CpG content, and these were used in most downstream enrichment testing to control for genomic context. We then examined the DMRs for each pairwise comparison for potential consistency across time and tissue. A large subset of the DMRs from the pairwise contrasts map to genes that overlap for obese vs. control (Fig. 1b), caloric restriction vs. obese (Fig. 1c), and pravastatin vs. obese (Fig. 1d). The overlaps are not only apparent across different pregnancy timepoints and brain regions for their respective sources but a subset also converge between cffDNA and brain. Next, the genomic coordinates of the

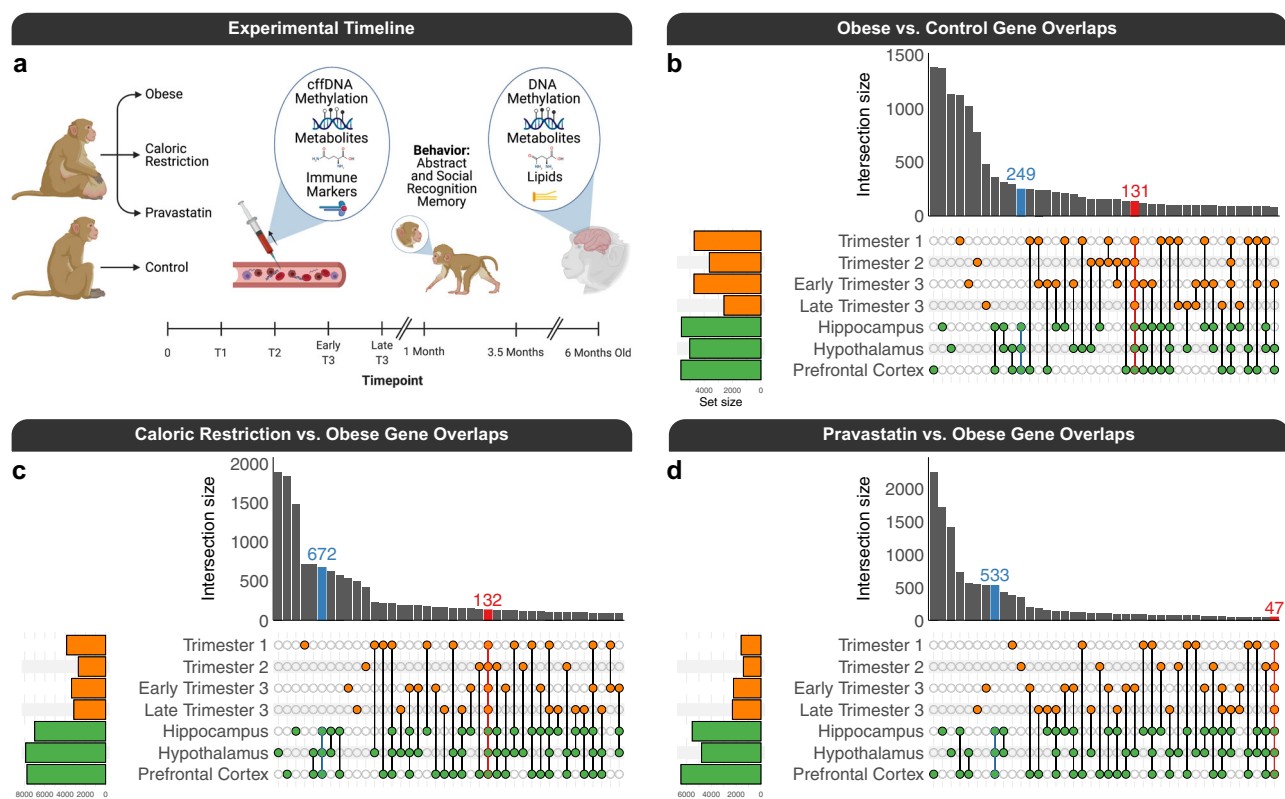

**Fig. 1 | cffDNA and brain DMR overlaps. a** Experimental design and timeline (created with BioRender.com). UpSet plots of the overlaps of gene mappings from pairwise DMR comparisons of **b** obese vs. control for cffDNA from trimester 1 ($n_{obese} = 7$, $n_{control} = 4$), trimester 2 ($n_{obese} = 7$, $n_{control} = 4$), early trimester 3 ($n_{obese} = 7$, $n_{control} = 5$), and late trimester 3 ($n_{obese} = 7$, $n_{control} = 6$) as well as infant brain ($n_{obese} = 7$, $n_{control} = 6$ for hippocampus, hypothalamus, and prefrontal cortex), **c** caloric restriction vs. obese for cffDNA from trimester 1 ($n_{caloric\ restriction} = 4$, $n_{obese} = 7$), trimester 2 ($n_{caloric\ restriction} = 5$, $n_{obese} = 7$), early trimester 3 ($n_{caloric\ restriction} = 5$, $n_{obese} = 7$), and late trimester 3 ($n_{caloric\ restriction} = 7$, $n_{obese} = 5$) as well as infant brain ($n_{caloric\ restriction} = 5$, $n_{obese} = 7$ for hippocampus, hypothalamus, and prefrontal cortex), and **d** pravastatin vs. obese for cffDNA from trimester 1 ($n_{pravastatin} = 7$, $n_{obese} = 7$), trimester 2 ($n_{pravastatin} = 7$, $n_{obese} = 7$), early trimester 3 ($n_{pravastatin} = 7$, $n_{obese} = 7$), and late trimester 3 ($n_{pravastatin} = 7$, $n_{obese} = 7$) as well as infant brain ($n_{pravastatin} = 7$, $n_{obese} = 7$ for hippocampus, hypothalamus, and prefrontal cortex). Source data are provided as a Source Data file.

DMRs, which were determined independently from each pairwise comparison, were merged into separate consensus regions for cffDNA and brain and the same was done for their respective background regions, also determined independently in each comparison. The overlap between cffDNA and brain consensus DMRs was significant (empirical $p = 0.0001$) in two separate analytical approaches, which included a permutation approach ($n = 10,000$) based on region overlap that placed the DMRs randomly across the entire genome, while maintaining their size, and a random sampling approach ($n = 10,000$) based on nucleotide overlap that utilized background regions with similar genomic context (CpG content and length).

The convergence of DNA methylation alterations associated with maternal obesity was consistent with the gene ontology (GO) analyses of the cffDNA (Fig. 2a) and brain (Fig. 2b) consensus DMRs, relative to their background regions, which were enriched ($q < 0.05$) for terms related to neurodevelopment, cellular adhesion, and cellular signaling. Additionally, the top GO terms from the cffDNA DMRs (anatomical structure morphogenesis, nervous system development, anatomical structure development, plasma membrane bounded cell projection organization, cell adhesion, and movement of cell or subcellular component) and the brain DMRs (nervous system development, cell junction, and cytoskeletal protein binding) also passed a more stringent significance (FWER < 0.1) threshold, which was based on 100 random sets from samplings of their respective consensus background regions. The GO terms were also consistent with the significant ($q < 0.05$) PANTHER (Protein Analysis THrough Evolutionary Relationships) pathways, which demonstrated a shared effect on integrin

signaling, glutamatergic synapses, and angiogenesis in both cffDNA and brain (Fig. 2c, d). Next, to examine the gene regulatory relevance of the consensus DMRs, they were tested for enrichment within human transcription factor motifs from a methylation-sensitive SELEX (Systematic Evolution of Ligands by EXponential enrichment) experiment[49]. The top significant ($E$ value < 0.05) transcription factors for both the cffDNA (Fig. 2e) and brain (Fig. 2f) consensus DMRs, relative to their background regions, were from the hairy and enhancer of split (HES) and activating protein-2 (AP-2) families, and overall the top enrichments were related to methylation-sensitive developmental transcription factors.

The relevance of the obesity consensus DMRs to previously identified human NDD-associated DMRs was tested by lifting over the consensus DMRs to the human genome (hg38). The lifted-over consensus DMRs were tested for enrichment within DMRs from updated analyses of previously published male idiopathic ASD brain, male Dup15q syndrome brain, female Rett syndrome brain, and male Down Syndrome brain and unpublished idiopathic male and female ASD placenta[31,50–52]. In a permutation approach based on region overlap, the consensus cffDNA and brain DMRs significantly ($q = 0.0012$) overlapped with all human NDD datasets. In a random sampling approach of background regions based on nucleotide overlap, the DMR overlaps showed similar significance ($q < 0.05$), although neither set of consensus DMRs showed a significant enrichment for the female ASD placenta DMRs (Table 1). Notably, the male Down syndrome brain DMRs had a lower enrichment score than the other ASD-related brain DMR datasets in both analytical approaches.

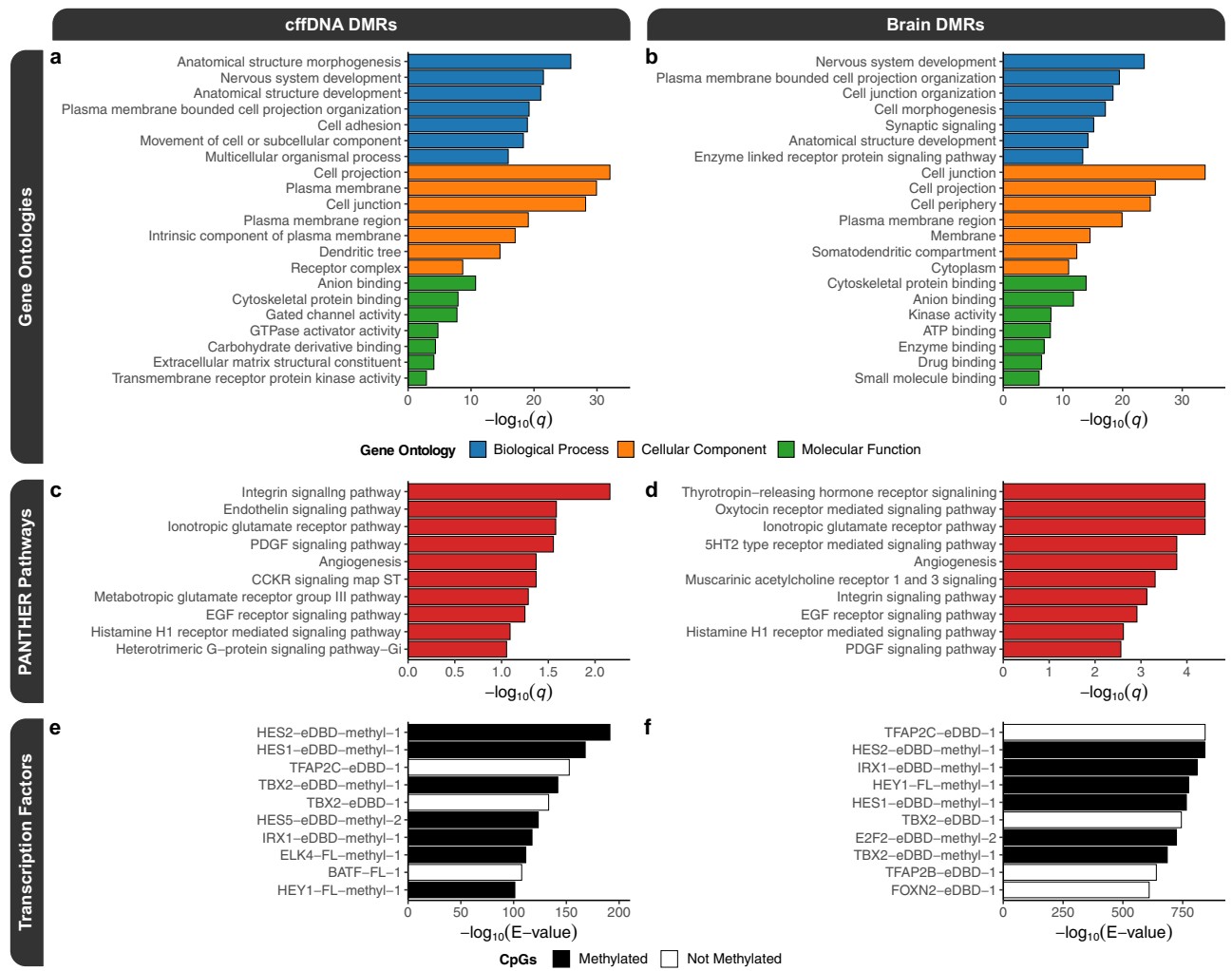

**Fig. 2 | Functional enrichments for the consensus cffDNA and brain DMRs.** Top slimmed gene ontology (GO) enrichment testing results for the **a** cffDNA and **b** brain consensus DMRs. Top PANTHER pathway enrichments for the **c** cffDNA, and **d** brain consensus DMRs. Top human methylation-sensitive transcription factor motif enrichment testing results for the **e** cffDNA and **f** brain consensus DMRs. The motif names indicate whether the transcription factor was full-length ("-FL") or an extended DNA-binding domain ("-eDBD"), if the CpGs were methylated ("-methyl"); and the number (starting with "−1") distinguishes between multiple motifs. Source data are provided as a Source Data file.

## A large block of DNA hypermethylation overlapping mir-663 is shared between cffDNA and brain

In addition to the DMR analyses, we also performed a separate analysis to detect larger-scale blocks of differential methylation in cffDNA and brain significantly associated with maternal obesity and intervention. The top overall hit in multiple pairwise contrasts were regions within a larger block of differential methylation (chr20:29790471-29824182, width = 33,712 bp) that was hypermethylated by maternal obesity in both cffDNA and brain. The block was primarily represented by a CpG dense 14.6 kb region (Fig. 3a). In cffDNA, the maternal obesity group showed the highest methylation level, the intervention groups showed an intermediate level of methylation, and the control group showed the lowest level of methylation. In brain, the maternal obesity group showed the highest level of methylation and the caloric restriction group showed the lowest. The block mapped to a cluster of genes that code for mir-663, ribosomal RNAs (28 S, 18 S, and 5.8 S), 2 novel lncRNAs, a pseudogene, and 3 novel protein coding genes, with mir-663 being the only well characterized gene. These results demonstrate a maternal obesity associated DNA methylation difference that spans an 33.7 kb chromosomal locus and is stable across time and tissue type.

While differentially methylated regions and blocks were discovered by group differences, we also performed correlation analyses of methylation levels of the block from individual animals with maternal group and assays of maternal blood across pregnancy and infant brain regions. The methylation levels for all cffDNA timepoints and brain regions showed positive correlations with maternal group (Fig. 3b-c). The longitudinal cffDNA methylation levels of the *mir-663* block also correlated with inflammatory maternal immune markers and distinct metabolites during their respective pregnancy timepoint (Fig. 3b). The infant brain region methylation levels of the block showed significant negative correlations with caloric restriction and abstract stimuli recognition memory as well as a significant positive correlation with social stimuli (face) recognition memory (Fig. 3c). The brain methylation values also correlated with distinct metabolites measured from their respective brain regions, and the prefrontal cortex showed a negative correlation with C20:3n-3 (Homo-γ-linolenic acid/8,11,14-eicosatrienoic acid) concentrations in that brain region. The observed group differences in methylation were consistent with pairwise comparisons of maternal group in a linear mixed effects model, with maternal group, timepoint or brain region, maternal age, cohort, birth GD, C-section, and foster status as fixed effects and individual as a random effect. Specifically, for cffDNA there was a trend ($p = 0.07$) for a difference between obese and control, and for brain there was a

**Table 1 | Human NDD-associated DMR enrichments for the consensus cffDNA DMRs and consensus brain DMRs**

| Dataset | | | cffDNA DMRs | | Brain DMRs | |
|---|---|---|---|---|---|---|
| NDD | Sex | Tissue | Fold | q value | Fold | q value |
| ASD | Female | Placenta | 1.05 | 0.3 | 1.08 | 0.1 |
| ASD | Male | Placenta | 1.25 | 0.04 | 1.26 | 0.03 |
| Down Syndrome | Male | Brain | 1.12 | 0.008 | 1.37 | 0.0002 |
| ASD | Male | Brain | 1.36 | 0.0002 | 1.51 | 0.0002 |
| Rett Syndrome | Female | Brain | 1.53 | 0.0002 | 1.69 | 0.0002 |
| Dup15q Syndrome | Male | Brain | 1.28 | 0.0002 | 1.74 | 0.0002 |

significant ($p = 0.01$) difference in methylation at this locus between obese and caloric restriction (Fig. 3d).

### A DUX4 co-methylation network in infant hippocampus correlates with maternal obesity, behavior, metabolites, lipids, and cffDNA methylation

In order to further investigate gene networks associated with maternal obesity and integrate with additional data sets, weighted gene co-methylation network analysis (WGCNA) of WGBS data from infant hippocampus was performed. The signed network demonstrated scale-free topology (Supplementary Fig. 2), leading to the identification of five modules of co-methylated regions and their respective hub regions (Table 2). The identified modules were tested for correlations with a suite of traits that included behavioral tests relevant to the hippocampus (abstract stimuli and social stimuli recognition memories) as well as lipidomic and metabolomic measurements from the hippocampus (Fig. 4a). Notably, the blue module showed a number of significant ($p < 0.05$) negative and positive correlations with all classes of traits measured. The blue module eigengene was negatively correlated with the maternal obesity group, positively correlated with the abstract stimuli recognition memory, and negatively correlated with the social stimuli (face) recognition memory (Fig. 4a, b). The blue module eigengene was also positively correlated with the concentration of two polyunsaturated essential fatty acids (PUFAs) in the hippocampus and linoleic acid was the most prominent (Fig. 4a, b). The blue module eigengene was also positively correlated with the concentrations of the metabolites asparagine and citrate in the hippocampus (Fig. 4a, b).

The blue module was composed of 104 regions out of 151,892 regions examined (Fig. 4c and Supplementary Data 3). On average, the regions in the blue module were 1841 bp and they mapped to 82 unique genes. The hub for the blue module was an intergenic region that mapped to ENSMMUG00000060367, which is a novel gene that is an ortholog for human *DUX4* (*double homeobox 4*) and is termed *DUX4* (region) 6. Notably, the blue module was composed of 21 inter-connected regions that mapped to *DUX4*, which spanned 206,145 bp (chr9:427125-633269), were between −158,396 bp to 44,716 bp from the *DUX4* transcription start site (TSS), and were hypomethylated overall in hippocampi from the offspring of obese dams. The regions in the *DUX4* co-methylated network mapped to genes that were primarily associated with functions related to gene regulation, metabolism, immunity and inflammation, NDDs, oxidative stress, obesity and adi-pogenesis, Wnt signaling, glutamatergic synapses, endoplasmic reti-culum, and reproduction (Table 3).

Next, the blue module eigengene was tested for correlations with traits from maternal blood across four time points that represented all trimesters of pregnancy, specifically with *DUX4* cffDNA methylation levels, cytokine levels, and metabolite levels (Fig. 4d). These relation-ships were dynamic across pregnancy, as cffDNA *DUX4* methylation levels within region 1 of the module (chr9:631319-633269, width = 1951 bp) were positively correlated with the blue module eigengene in

trimester 1 but negatively correlated with the module in early trimester 3. There were strong negative correlations between the blue module eigengene and levels of several immunological indicators of obesity associated inflammation, with the strongest association being MCP-1 (CCL2) levels during trimester 1, IL-10 during trimester 2, sCD40L during early trimester 3. Several metabolites in maternal blood throughout pregnancy showed both positive and negative correlations with the blue module eigengene, including a positive correlation with creatine during trimester 1 and 2, as well as negative correlations with glutamine and arginine during trimester 1. The maternal metabolites showing the most prominent correlations with the hippocampal blue module eigengene were generally related to one-carbon metabolism (choline, creatine, and glycine) and metabolism of amino acids by the tricarboxylic acid cycle ($\alpha$-ketoglutaric Acid, arginine, glutamine, and succinate), which is also known as the citric acid cycle and the Krebs cycle[53,54].

## Discussion

There are four key findings from this integrative multi-omic analysis of offspring DNA methylation and outcomes resulting from exposure to maternal obesity and intervention in a non-human primate model that are relevant to human NDD/ASD. First, the longitudinal analysis of cffDNA throughout pregnancy demonstrated that cffDNA methylation was consistent with the functions and pathways disrupted in infant brain, particularly for DNA methylation patterns over large genomic blocks at *mir-663* and *DUX4*. Second, the caloric restriction and pra-vastatin intervention groups displayed intermediate methylation levels in these regions throughout pregnancy. Third, cffDNA methy-lation levels at *DUX4* correlated with the co-methylated *DUX4* gene network in 6-month infant hippocampus, which correlated with infant social and abstract recognition memory, infant hippocampal and maternal blood metabolites, and infant hippocampal lipids. Fourth, the maternal obesity DMRs overlapped with DMRs from human ASD/NDD brain and placenta.

To expand on the above summary, the results demonstrate that maternal obesity is associated with the differential methylation of a subset of common genes in both cffDNA and brain. The differentially methylated genes associated with maternal obesity in cffDNA and brain are enriched for functions related to neurodevelopment, cellular adhesion, and cellular signaling. Notably, they converge on pathways known to be affected in NDD/ASD. These include glutamatergic synapses[55], angiogenesis[56], integrin signaling (which is involved in cellular adhesion)[57], EGF receptor signaling, and PDGF signaling. Additionally, the brain displayed differences in methylation that were consistent with other neurotransmitters and hormone signaling pathways related to NDDs/ASD, which include thyrotropin (also known as thyroid stimulating hormone; TSH), oxytocin[58,59], serotonin, and acetylcholine.

Functionally, the DMRs are consistent with disruptions to the regulation of gene expression, since they are enriched for the motifs of human transcription factors that are methylation-sensitive and involved in early development[49]. The top motif enrichments overall were for the HES and AP-2 families. The *HES* genes are transcriptional repressors involved in early embryonic development and neurodeve-lopment that function to regulate the differentiation and proliferation of neural stem cells[60,61]. The binding of HES transcription factors to their motifs is inhibited by DNA methylation[49]. The *HES* genes are effectors of the Notch signaling pathway and cross-talk with JAK-STAT signaling, which is activated by cytokines[62]. Additionally, *HES1* is a thyroid response gene in the fetal brain[63], which was the top pathway for the fetal brain DMRs associated with maternal obesity and inter-vention in our study. The AP-2 transcription factors are also involved in early embryonic development, where they stimulate cell-type specific proliferation and repress terminal differentiation[64]. The top motif from this family belongs to transcription factor gene AP-2 gamma (TFAP2C),

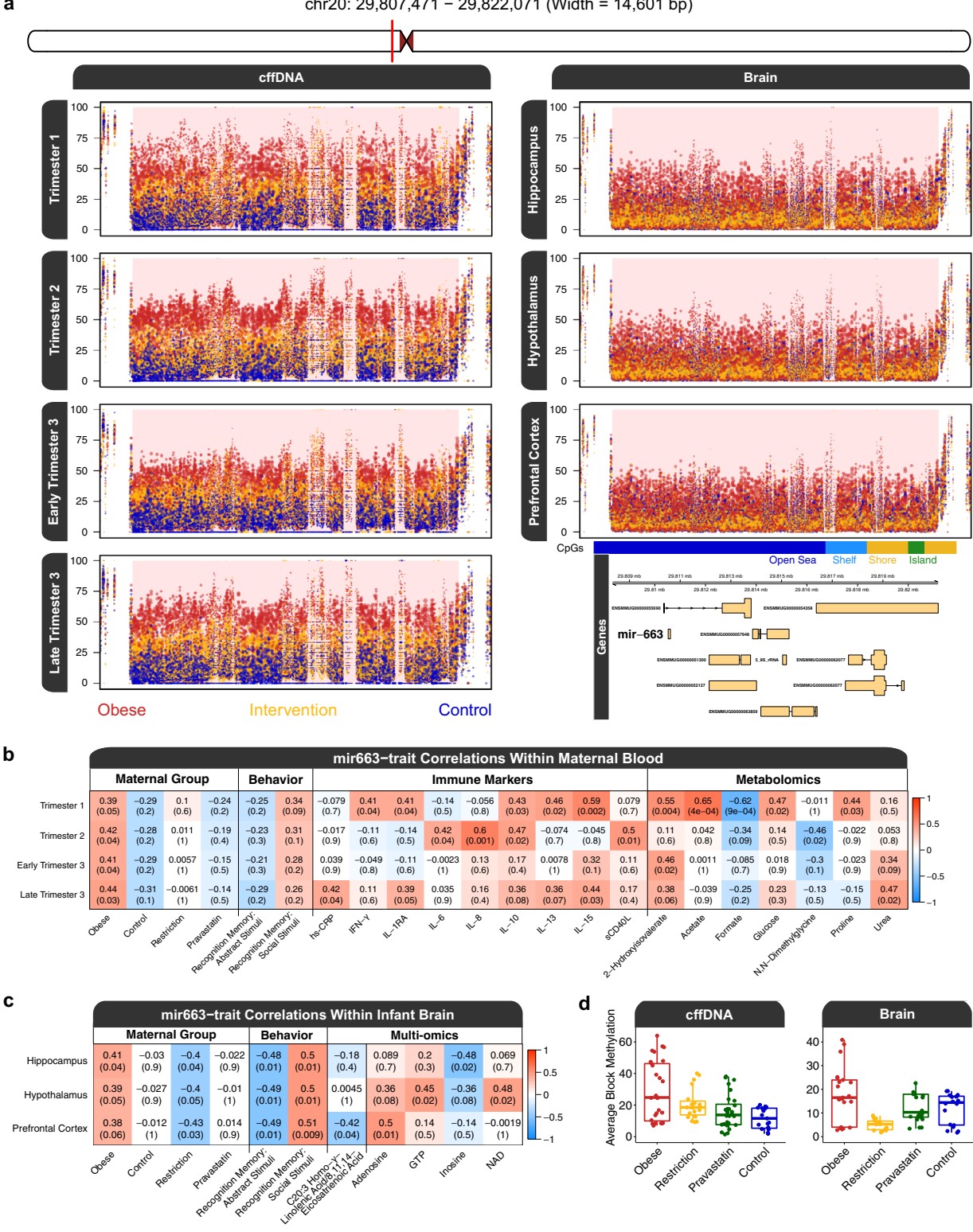

which is specifically involved in both placental development and redundantly for retinoic acid induced differentiation of the neural tube[65,66], and thus represents a direct connection between the two sample sources.

In addition to discovering a profile of thousands of DMRs of several hundred bp in width, in cffDNA and brain, we also identified larger-scale blocks of differential methylation in cffDNA and brain that associated with maternal obesity and intervention. The mir-663 block was ~34 kb in size and overlapped several genes, many of which have

unknown functions and warrant future functional research. However, mir-663 is associated with obesity and adipocyte differentiation, immunity and inflammation, the mechanism of resveratrol action, and cancer in humans[67–71]. Elevated mir-663 levels were also observed in ASD compared to control lymphoblastoid lines in a small study[72]. In our study, hypermethylation of the block in cffDNA correlated with maternal obesity as well as increased maternal inflammatory markers and differential metabolite levels during pregnancy. In infant brain, hypermethylation of the block not only correlated with maternal

**Fig. 3 | A large block of obesity-associated DMR hypermethylation in cffDNA and brain.** The plots and statistical testing are based on the region of highest CpG density (chr20:29807471-29822071, width = 14,601 bp), which represents the primary signal of the entire block (chr20:29790471-29824182, width = 33,712 bp). **a** Plot of methylation levels in the main block region. The dots in the scatter plot are individual DNA methylation level estimates for a CpG site and their size reflects the level of coverage from the sequencing. Percent DNA methylation is presented on the *y* axis and the *x* axis is the genomic coordinate of each CpG, where the ticks show the location of the methylation loci. The bottom right track contains CpG annotations and gene mappings for the block. **b** Correlation heatmap of the relationship between longitudinal cffDNA methylation levels in the block and maternal blood immune markers and metabolites measured during the same trimester that showed a significant correlation (*p* < 0.05) within at least one time point. Pearson's correlation coefficients (*r*) are reported above their *p*-values, which are in parentheses. The obesity group refers to maternal obesity with no intervention. The heatmap colors are representative of the correlation between the methylation values and the trait of interest. **c** Correlation heatmap of the relationship between infant brain region DNA methylation levels in the block and infant hippocampal lipids and metabolites measured from the same brain region that showed a significant correlation (*p* < 0.05) within at least one brain region. **d** Average smoothed methylation levels for cffDNA and brain. The center line of the boxplot represents the median and the bounds represent the interquartile range (IQR), which is between the 25th percentile (Q1) and 75th percentile (Q3). The whiskers of the boxplot extend to the maxima (Q3 + 1.5 × IQR) and minima (Q1−1.5 × IQR). Source data are provided as a Source Data file.

### Table 2 | WGCNA module hub genes

| Module | Coordinates | Width | Annotation | Gene mapping |
|---|---|---|---|---|
| Green | chr18:45038579-45038736 | 158 | Intron | *CELF4*<br>*CUGBP Elav-like Family Member 4*<br>ENSMMUG00000006866 |
| Yellow | chr4:139959607-139965208 | 5602 | 3' UTR | *MAMU-A3*<br>*Major Histocompatibility Complex, Class I, A*<br>ENSMMUG00000056914 |
| Blue | chr9:443230-472517 | 29288 | Intergenic | *DUX4*<br>*Double Homeobox 4*<br>ENSMMUG00000060367 |
| Brown | chr10:26959768-26960135 | 368 | Intergenic | ENSMMUG00000051997 |
| Turquoise | chrY:8116120-8122200 | 6081 | Intergenic | *LOC106995433*<br>*Heat Shock Transcription Factor, Y-linked-like*<br>ENSMMUG00000049379 |

obesity and infant metabolites, but also with infant social and abstract recognition memory, but negatively with the caloric restriction intervention. The effect on brain lipids was most pronounced in the prefrontal cortex and through decreased concentrations of C20:3n-3 (Homo-γ-linolenic acid/8,11,14-eicosatrienoic acid), which has anti-inflammatory effects. Through a WGCNA approach, we also identified a key co-methylated network whose hub was a large ~200 kb block of differential methylation that mapped to *DUX4*. *DUX4* is a homeobox transcription factor that is expressed in cleavage stage embryos and testes, and is epigenetically silenced in most other tissues[73]. Incomplete silencing of *DUX4*, which is located in the D4Z4 repeat in humans, results in Facioscapulohumeral muscular dystrophy (FSHD) through pathogenic misexpression of *DUX4* in skeletal muscle due to DNA hypomethylation of the locus[74]. This misexpression ultimately leads to an immune deregulation cascade[73], and can be repressed by targeted epigenetic editing[75]. Notably, the *DUX4* DNA hypomethylation in FSHD directionally corresponds with the *DUX4* co-methylation network we observed in hippocampus, where the maternal obesity with no intervention group was hypomethylated when compared to the control and obesity intervention groups.

The regions in the *DUX4* co-methylation network mapped to genes with functions highly related to its significantly correlated phenotypes. The *ERBB2 (Erb-B2 Receptor Tyrosine Kinase 2)* mapping is consistent with the differences in recognition memory as the gene is known to regulate hippocampal glutamatergic long-term depression and object recognition memory[76]. Additionally, the glutamatergic synapse is represented in the network by *OLFM3* and *PRKX*, as well as in the top pathways for the consensus cffDNA and brain DMRs. The top pathway from the brain DMRs is also represented in the co-methylation network by *SLC16A2*, which is a thyroid hormone transporter. Of relevance to the lipidomic profile is a region mapping to *FFAR4 (Free Fatty Acid Receptor 4)*, which is a GPCR (*GPR120*) for PUFAs that is involved in adipogenesis, metabolism, and inflammation[77,78]. Levels of linoleic acid (LA, 18:2n-6), an omega-6 PUFA that is a ligand

for FFAR4, correlated with the co-methylation network. LA is known to increase neurite outgrowth in the developing brain[79–81], and lower levels of LA have also been observed in the serum of children with autism[82]. The impact of the *DUX4* co-methylation network on neurodevelopment is also apparent through the correlation of asparagine, as the network contains the gene *ASNS (Asparagine Synthetase)*. *ASNS* deficiency is a neurometabolic disorder characterized by severe congenital microcephaly and developmental delay[83]. Other notable genes in the co-methylation network include: *PCDH11X*, which is associated with ASD[84], *ZFHX3* and *ZFHX4*, which are homeobox genes that act as transcription factors that regulate myogenic and neuronal differentiation, and *ASIP (Agouti Signaling Protein)*, which is involved in obesity[85].

The *DUX4* co-methylation network also showed several significant negative correlations with maternal blood markers during pregnancy that are known to be associated with obesity. Methylation of the *DUX4* module was negatively correlated with inflammatory cytokines/adipokines in maternal blood throughout different trimesters of pregnancy, including MCP-1 (CCL2), which is both a monokine and adipokine[86], as well as the chemokine IL-8 and the regulatory cytokine IL-10, which are associated with excess bodyweight[87]. Increased levels of MCP-1 and IL-8, have been observed in the blood of human mothers with obesity during pregnancy[17,88]. Finally, the differences in maternal metabolites are consistent with previously known impacts on one-carbon metabolism[14–16].

Taken together, the correlations of the *DUX4* co-methylation network in infant hippocampus demonstrate that the offspring from lean control and intervention groups had higher levels of methylation than those from obese dams in the *DUX4* co-methylation network, which significantly correlated with differences in behavior related to abstract stimuli and social stimuli recognition memory, higher PUFA concentrations in the brain, and differing levels of metabolites related to neurodevelopment. Furthermore, the higher methylation levels of this module significantly correlated with lower levels of maternal

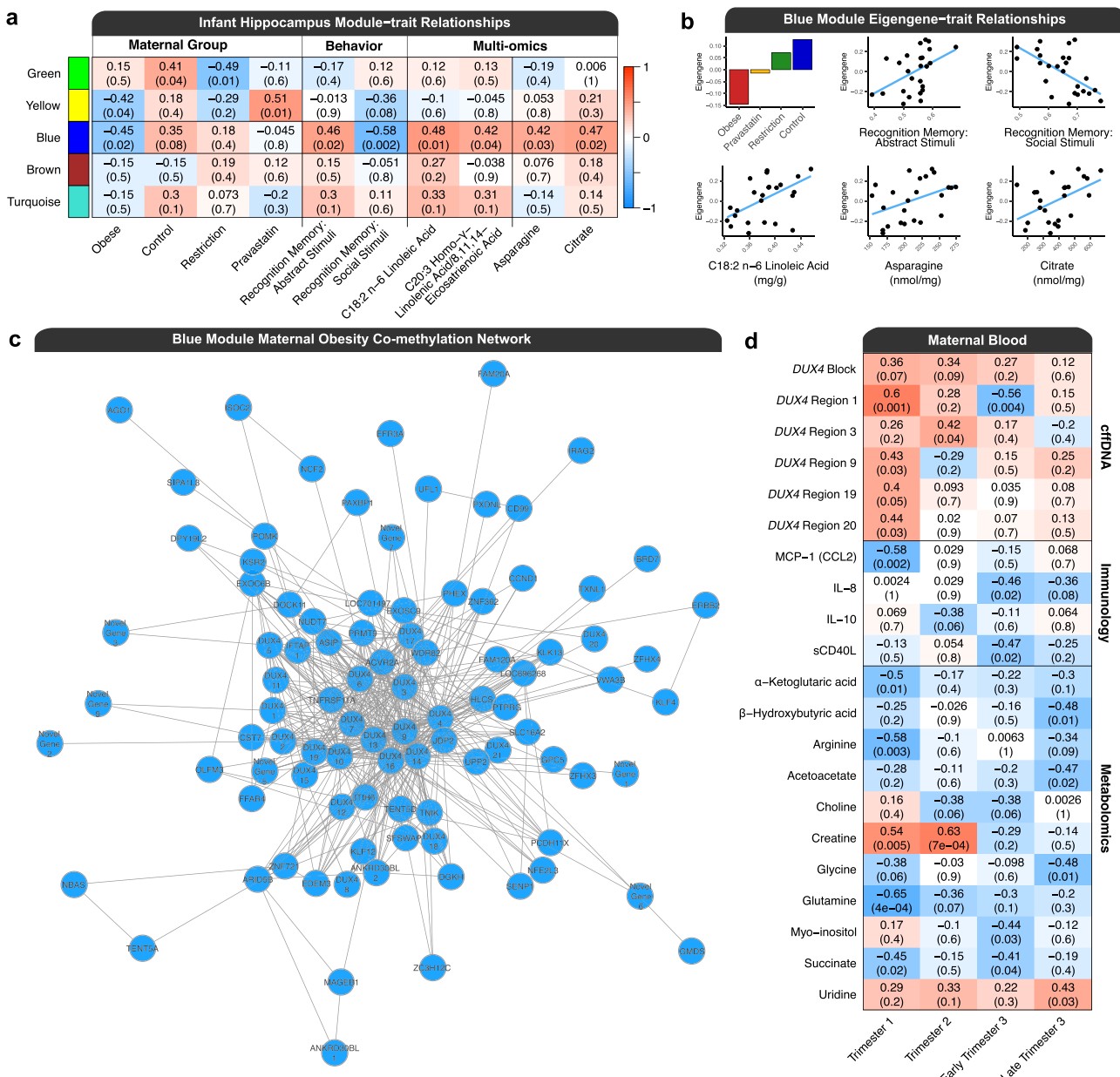

**Fig. 4 | Weighted gene co-methylation network (WGCNA) of infant hippocampus. a** Module-trait correlations within the hippocampus. The heatmap colors are representative of the correlation between the module eigengenes and the trait of interest. The obesity group refers to maternal obesity with no intervention. Pearson's correlation coefficients (*r*) are reported above their *p* values, which are in parentheses, and these values also apply to the plots in part B of the figure. **b** Bar plot of the mean eigengene values for the blue module in each maternal group, and scatter plots of all animals across all four groups with a line of best fit for the eigengene values and abstract stimuli recognition memory score ratios, social stimuli recognition memory score ratios, linoleic acid concentrations, asparagine concentrations, and citrate concentrations. **c** The blue module maternal obesity comethylation network. Regions were mapped to their nearest gene and novel genes were labeled with mammalian ortholog symbols, if available. Genes represented by more than one region are appended with a unique number identifier. Edges were included in the network if they passed an adjacency threshold and thus not all genes in the module are represented in the visualization. **d** Hippocampal blue module-trait correlations with maternal blood measurements of *DUX4* cffDNA methylation levels, immunological markers, and metabolites across all trimesters of pregnancy. Source data are provided as a Source Data file.

inflammatory cytokines/adipokines and differences in maternal one-carbon metabolism and metabolism of amino acids by the tricarboxylic acid cycle during pregnancy.

A potential limitation of this study is the small sample size characteristic of all NHP studies, especially perinatal studies[89]. However, we controlled for sex (male offspring only) and used a larger sample size than most similar NHP studies. The unique insights that our study has provided to the field would not have been ethically possible in a human study and the identification of two genomic loci (*mir-663* and *DUX4*) that have primate-specific genomic organization and regulation, along

with the NDD/ASD relevant recognition memory assays, would not have been possible in a rodent study. While the main conclusions of our study are based on large consistent changes in methylome signatures over multiple or broad genomic regions, another potential limitation is the lower confidence associated with any individual DMR reported, which would require replication. Finally, the DNA methylation correlation analyses were able to identify significant relationships in behavioral, metabolomic, immunological, and lipidomic data that did not show significant group differences in isolation. This multi-omic correlation based approach appears well suited for this biological

**Table 3 | Manually curated blue module co-methylated network gene mapping categories**

| Category | Genes |
| --- | --- |
| Gene regulation | *AGO1, ARID5B, BRD7, DUX4, ETV1, EXOSC9, JDP2, HLCS, KLF4, KLF12, NFE2L3, PAXBP1, PRMT9, SFSWAP, WDR82, ZC3H12C, ZFHX3, ZFHX4, ZNF362, ZNF721* |
| Metabolism | *ASNS, DGKH, DPY19L2, EDEM3, EXOSC9, FFAR4, GMDS, HLCS, ITIH6, KLF4, KLK13, LYVE1, NUDT7, PHEX, PRMT9, SENP1, UPP2* |
| Immunity and inflammation | *ARID5B, ACVR2A, CD99, CST7, DOCK11, EXOSC9, FFAR4, IFTAP, IRAG2, LRRC32, LYVE1, NCF2, PRKX, TNFRSF11A, ZC3H12C* |
| Neurodevelopmental disorders | *ASNS, CHL1, EFR3A, ERBB2, EXOC6B, GPC5, LRRC32, PCDH11X, POMK, PTPRG, SLC16A12, TNIK, VWA3B, WDR82* |
| Oxidative stress | *FAM120A, NCF2, NFE2L3, NUDT7, PXDNL, TXNL1* |
| Obesity and adipogenesis | *ACVR2A, ARID5B, ASIP, FFAR4, KLF4, KSR2* |
| Wnt signaling | *BRD7, JDP2, PCDH11X, SHISA7, TNIK* |
| Glutamatergic synapses | *ERBB2, OLFM3, PRKX* |
| Endoplasmic reticulum | *EDEM3, IRAG2, NFE2L3, UFL1* |
| Reproduction | *DPY19L2, MAGEB1, MROH5* |

question since not every human child born to a mother with obesity develops a NDD.

Ultimately, the methylation profiles of both the *mir-663* block and the *DUX4* co-methylation network from the infants of obese dams without an intervention correlated with decreased recognition memory for abstract stimuli and increased recognition memory for novel social stimuli (faces). The findings demonstrate that, in infant brain, maternal obesity is associated with a DNA methylation profile at gene loci relevant to recognition memory, lipids, and metabolites. These differences in brain multi-omics and behavior can be detected during pregnancy through integrative analyses of cffDNA with immune and metabolic factors. Furthermore, maternal obesity interventions associated with an attenuation of the multi-omic profile in both infant brain and maternal blood.

## Methods

### Non-human primate obesity models

All animals were housed at the California National Primate Research Center (CNPRC) in accordance with the ethics guidelines established and approved by the Institutional Animal Use and Care Administrative Advisory Committee at the University of California-Davis.

Adult pregnant female rhesus macaques (*Macaca mulatta*) with male fetuses were selected for this study. Sex of the fetus was determined with a cffDNA Y chromosome gene analysis of maternal blood and was performed early in the first trimester by the CNPRC Primate Assay Core. All dams ranged in age from 7 to 12 years and were selected for lean and obese groups based on their Body Condition Score (BCS)[90]. Obese females had a BCS of at least 3.5 (range 1–5) which correlates with 32% body fat, and lean animals had a BCS of 2–2.5. Animals had maintained a consistent BCS for at least one year prior to selection for the study and pre-study physicals confirmed that none of the selected females were diabetic. All animals were maintained with standard indoor housing conditions at CNPRC and fed nine "biscuits" of commercial chow (High Protein Primate Diet Jumbo; LabDiet; 5047) twice daily while pregnant, received biweekly fresh produce, daily forage mixture, and *ad libitum* water. The caloric restriction group had the amount of chow restricted to prevent weight gain during pregnancy and the Pravastatin group was given 1 mg/kg of body weight. However, all dams, regardless of group, were provided twelve biscuits twice daily during nursing of 4 months or older infants. Dams were relocated to a single housing room around gestational day 70. Approximately 2 weeks later they were paired with a compatible cage mate during daytime hours and were separated prior to feeding times. Dams were allowed to deliver naturally (~165 day gestation length) and mother-infant pairs were raised indoors until offspring were 6 months old. However, five pregnant dams required Cesarian deliveries for post-date pregnancies (~175 gestation days) or as recommended by veterinarians for health reasons, two in the obese group, one in the

pravastatin group and two in the lean control group. In those scenarios, infants were successfully reared to 6 months of age by foster dams using established CNPRC protocols[91]. Mother-infant dyads were housed indoors with another compatible mother-infant dyad during daytime hours if possible; however, this was not possible for 1 lean control dyad, 3 obese dyads, and 1 pravastatin dyad. Infant brain samples (hippocampus, hypothalamus, and prefrontal cortex) were collected on postnatal day 180 after infants were anesthetized with ketamine and euthanized with 120 mg/kg pentobarbital. Upon collection, samples were immediately frozen and stored at −80 °C.

### Abstract stimuli recognition memory

Infants were tested at 200 days gestational age (days from conception; ~1-month old) to avoid differential maturity due to variation in gestation length (range: 152–176 days). The mean postnatal age was 36 days (range: 22–49 days). Infants were separated from mothers, wrapped in a towel, and carried to the testing station. The testing apparatus consisted of a small booth with side panels to shield from outside distractions in a darkened room in which the stimuli, 9 cm$^2$, were mounted on the left and right of a center viewing hole. One tester held the infant 36 cm away from the stimuli, changed the stimuli, and covered the infant's head between trials. A second tester sat behind the apparatus and viewed the infant's head through a video camera to record fixation times to the left and right. Two identical stimuli were placed on the right and left until the infant accumulated a familiarization time of 20 sec. One stimulus, randomly determined, was then removed and replaced with the novel stimulus. The frequency and duration of looking were recorded for 10 sec, then the stimuli were exchanged between sides for another 10 sec test period. The infant was presented with a series of four black and white visual stimuli pairs (abstract illustrations of varying complexity) used in the Fagan Test of Infant Intelligence for human infants (Infantest Corp). Videos were scored for right/left looking using The Observer coding software (version 12.5, Noldus) by a coder blind to stimulus location and animal treatment group. The evaluation relies on the propensity of monkey as well as human infants to look longer at novel than familiar visual stimuli. The outcome measure used in this study was the average across the four problems of the number of looks at the novel stimulus divided by the number of looks at the novel and familiar stimuli during test trials[92]. All infants in the lean control group had a ratio >0.50 indicating novelty preference.

### Social stimuli recognition memory

Animals were tested at a mean age of 105.3 days (range: 96-124). They were separated from their dams and relocated to individual housing indoors in a standard size housing cage (0.58 m × 0.66 m × 0.81 m, Lab Products). Approximately 2.5 h after the separation/relocation, they were given a visual paired comparisons task. Each animal was hand-

carried to a test cage measuring 0.387 m × 0.413 m × 0.464 m that was positioned 0.686 m from a 0.813 m monitor (Panasonic, KV 32540), was given 30-sec to habituate, and was then presented with seven problems from a pre-recorded video. Each problem included three trials: a familiarization trial and two recognition trials. After a 5-sec blank screen, a 20-sec familiarization trial began, in which two identical pictures were presented, each measuring 19.7 × 22.9 cm, separated by 25.4 cm of white space onscreen. After another 5-sec delay, an 8-sec recognition trial occurred, in which the now-familiar stimulus was presented simultaneously with a novel stimulus (side determined randomly). Following another 5-sec delay, the same two stimuli were presented again for 8-sec, with positions reversed. Seven such problems were presented. All stimuli were pictures of unfamiliar juvenile and adult monkeys of both sexes[93]. A tone of 1000 Hz was presented 250 milliseconds prior to trials in order to orient the animal. A low-light camera (KT&C, KTLCMB5010EX), attached to the display monitor and situated midway between the two projected images, was used to record the subjects' looking responses. Looking behavior during the familiarization and recognition trials was scored by a trained observer who was blind to the animals' treatment groups. For each problem, the proportion of looking time directed at the novel stimulus was computed: duration of viewing the novel stimulus on the two recognition trials divided by the duration of viewing both the novel and familiar stimuli in the recognition trials. The principal outcome measure was a mean of this proportion across the seven problems. Chance responding was indicated by a mean of 0.50, with lower values suggesting a preference for the familiar stimuli and higher values indicating preference for the novel stimulus. Upon completion of testing, the subject was returned to its holding cage, and the test area was cleaned and prepared for the next subject.

## Lipidomics

The lipid extraction protocol was performed as previously described[94]. Briefly, 30 mg of homogenized hippocampus, hypothalamus, and prefrontal cortex samples were collection from matched regions. An optimized extraction protocol was used to get lipid and polar metabolites from small amounts of tissue, previously described using pig brain[94]. Upon obtaining Folch bottom layers, lipids were evaporated under nitrogen, reconstituted into 1.5 mL chloroform (Fisher Scientific; Cat #C607-4): isopropyl alcohol (Fischer Scientific, Cat #464-1) (v/v; 2/1). Approximately 0.5 mL of extract, with 0.0125 mg 5α-Cholestane (Sigma-Aldrich, Cat #C8003-100mg), and 0.1 mg C17:0 PC (1,2-diheptadecanoyl-sn-glycero-3-phosphocholine (Avanti Polar Lipids; 850355 C) was dried under nitrogen for fatty acid and cholesterol analysis. Upon drying, 400 μL of toluene (Fisher Scientific; T2914) was added, followed by 3 mL of methanol (Fisher Scientific; Cat #A454-4), and 600 μL of 3% HCl (Sigma-Aldrich; 320331) in methanol. The transesterification reaction to generate fatty acid methyl esters (FAMEs) was adapted from previous research[95]. Samples were vortexed and heated at 90 °C for 60 min. After cooling the samples at room temperature for 4–5 min, 1 mL of hexane (Fisher Scientific; H303-4) followed by 1 mL of deionized water was added to each sample. Samples were vortexed and the phases were allowed to separate for 15 min. Then, 900 μL from the hexane top layer containing FAMEs was transferred to microfuge tubes containing 450 μL of deionized water. The tubes were vortexed and centrifuged at 15,871 × g for 2 min. The top hexane layer was evaporated under nitrogen and then reconstituted in 100 μL hexane for GC-FID (Gas Chromatography with Flame-Ionization Detection) analysis.

A simultaneous FAME and cholesterol GC-FID method was developed by optimizing two previous methods[96,97]. Specifically, the LOQ is at least 10 times the signal to noise ratio for each identified FAME and the equation mass of FA/mass of internal standard C17:0 PC = area of FA/area of C17:0 was used to calculate each FA

concentration. A comparative study of GC-FID vs GC-MS has shown they yield similar results[98]. Samples were analyzed on a Perkin Elmer Clarus 500 GC-FID system (Perkin Elmer) equipped with a DB-FFAP polyethylene glycol fused capillary column (30 m × 0.25 mm inner diameter, 0.25 μm film thickness; Agilent Technologies; 1223232). The injector and detector temperatures were 285 °C and 300 °C, respectively. The initial oven temperature was 80 °C. It was held at 80 °C for 2 min, increased by 10 °C/min to 185 °C, raised to 249 °C at 6 °C/min and lastly held at 249 °C for 44 min. The total run time was 65 min. Helium was used as the carrier gas with a maintained flow rate of 1.3 mL/min. The injection volume was 1 μL per sample. The split ratio was 10:1. Each fatty acid was identified based on retention time using a custom-made mix of 29 FAME standards (C8:0, C10:0, C11:0, C12:0, C13:0, C14:0, C14:1, C15:0, C16:0, C16:1, C17:0, C18:0, C18:1, linoleic acid (C18:2 n-6), C18:3 n-6, alpha-linolenic acid (C18:3 n-3), C20:0, C20:1 n-9, C20:2 n-6, dihomo-γ-linolenic acid (C20:3 −6), arachidonic acid (C20:4 n-6), C20:3 n-3, eicosapentaenoic acid (C20:5 n-3), C22:0, C22:1, C22:2, n-6 docosapentaenoic acid (C22:5 n-6), n-3 docosapentaenoic acid (C22:5 n-3), docosahexaenoic acid (C22:6 n-3)/C24:1; Supplementary Fig. 3). Fatty acids were quantified using 1,2-diheptadecanoyl-sn-glycero-3-phosphocholine as an internal standard. Cholesterol concentration was calculated in a calibration curve using 5α-Cholestane as a surrogate. The linear dynamic range of the calibration curve for cholesterol was 0.25 mg/mL to 2 mg/mL. All chromatograms were analyzed on Totalchrom Navigator Software (version 6.3.2, PerkinElmer).

## Metabolomics

Fasting blood was collected in lavender top (EDTA) tubes (Cardinal Health; 8881311446) from mothers once during the 1st and 2nd trimesters and twice during the 3rd trimester after anesthetization with 5–30 mg/kg ketamine or 5–8 mg/kg telazol. Plasma samples were filtered using Amicon Ultra Centrifugal Filters (Millipore; UFC5003) to remove proteins and lipids. Metabolites were extracted from each of the brain tissues as previously described[94]. NMR metabolomics methods have been previously validated and published[99–101]. NMR-based metabolomics has shown high repeatability and reproducibility with low CV's of technical replicates, as well as low %error in reported concentrations[101]. The concentrations of individual metabolites are determined based on the addition of an internal standard as described in the methods. LOD/LOQ vary depending on the metabolite and sample matrix, but is generally between 1 mM and 50 mM, with most metabolites being able to reliably measured between 1 and 10 mM[99].

To 207 μL of either plasma filtrate or brain tissue extract, 23 μL of internal standard containing DSS-d6 was added and samples were placed in 3 mm Bruker nuclear magnetic resonance (NMR) tubes (Millipore; WIMWG30004SJ). Proton NMR spectra were acquired on each sample at 25 °C using the noesypr1d pulse sequence on a Bruker Avance 600 MHz NMR spectrometer (Bruker) and analyzed using Chenomx NMRSuite (version 8.1, Chenomx Inc) to annotated the metabolites based on Reference Library 10 as previously described[94].

## Immunology

A longitudinal analysis on the maternal cytokine/chemokine profile that included 22 analytes (GM-CSF, IFN-γ, IL-1b, IL-ra, IL-2, IL-4, IL-5, IL-6, IL-8, IL-10, IL-12/23(p40), IL-13, IL-15, IL-17a, IL-18, MCP-1, MIP-1b, MIP-1a, sCD40L, TGFα, TNFα, and VEGF) was measured in plasma using a non-human primate multiplexing bead immunoassay (Millipore-Sigma,; PCYTMG-40K-PX23) according to the manufacturer's protocol. The plates were read on a Bio-Plex 200 system (Bio-Rad Laboratories; 171000205) and analyzed using Bio-Plex Manager software (version 6.1, Bio-Rad Laboratories). A five-parameter curve was used to calculate final concentrations (pg/ml). Reference samples were run on each plate for assay consistency.

## DNA extraction and WGBS library preparation

The cffDNA was extracted from serum using a Maxwell RSC cffDNA Plasma Kit (Promega; AS1480) by the Primate Assay Laboratory Core at the California National Primate Research Center. The brain DNA was isolated from tissue stored in DNA/RNA shield (Zymo Research; R1100-250) using the Quick-DNA Miniprep Plus kit workflow on a Tecan instrument by Zymo Research. Brain DNA was fragmented using a E220 focused-ultrasonicator (Covaris; 500239). DNA was bisulfite converted using the EZ DNA Methylation-Lightning Kit (Zymo Research; D5031). WGBS library preparation was performed via the Accel-NGS Methyl-Seq DNA Library Kit (Swift Biosciences; 30096) with the Methyl-Seq Combinatorial Dual Indexing Kit (Swift Biosciences; 38096) according to the manufacturer's instructions. The primary cffDNA libraries were prepared by Swift Biosciences and the brain libraries were prepared by the UC Davis Genome Center. The primary cffDNA and brain library pools were sequenced by the UCSF Center for Advanced Technology (CAT) core facility on the Illumina NovaSeq 6000 S4 for 150 bp paired end reads. The pilot cffDNA library pool utilized the Methyl-Seq Set A Indexing Kit (Swift Biosciences; 36024) and was sequenced by the DNA Technologies and Expression Analysis Cores at the UC Davis Genome Center on an Illumina HiSeq 4000 for 90 bp single reads.

## Bioinformatic analyses

The CpG_Me alignment pipeline (https://github.com/ben-laufer/CpG_Me & https://doi.org/10.5281/zenodo.5030083), which is based on Trim Galore, FastQ Screen, Bismark, Picard, and MultiQC, was used to trim adapters and methylation bias, screen for contaminating genomes, align to the reference genome (rheMac10), remove duplicates, calculate coverage and insert size metrics, extract CpG methylation values, generate genome-wide cytosine reports (CpG count matrices), and examine quality control metrics[102–108]. cffDNA samples were examined for their ratio of chrY/chrX reads (https://github.com/hyeyeon-hwang/SexChecker).

DMR and block calling and most downstream analyses and visualizations were performed via DMRichR (https://github.com/ben-laufer/DMRichR & https://doi.org/10.5281/zenodo.5030057), which utilizes the dmrseq and bsseq algorithms[108–110]. In the dmrseq algorithm, differences in CpG methylation for the groups are pooled and smoothed to assemble background regions. To account for inter-CpG and inter-individual variability, a statistic for each region is estimated through generalized least squares regression and permutation testing is used to identify DMRs. A methylation difference of 10% was used for the cffDNA DMR analyses and 5% was used for the brain DMR analyses, as previously described for placenta and brain[31,51,52]. Background regions with similar genomic context to the DMRs (gene length and CpG content) were obtained from the first step of dmrseq and utilized in downstream enrichment testing. Consensus DMRs and background regions were assembled through merging the respective regions from each sample source-specific pairwise contrast by genomic coordinate overlap. Large-scale blocks of differential methylation were called in separate analyses using the blocks argument and recommended modifications to the smoothing parameters in dmrseq. ChIPseeker (was used to obtain gene region annotations and gene symbol mappings through ensembldb[111,112]. ComplexUpset was used to create UpSet plots of gene overlaps[113–115]. GOfuncR was used for genomic coordinate based GO analyses, where DMRs were mapped to genes if they were between 5 kb upstream to 1 kb downstream of the gene body, and 100 random samplings from the background regions with gene length correction was utilized for the enrichment testing[116,117]. The DMRs were mapped to genes if they were within 5 kb upstream or 1 kb downstream of the gene body. Redundant GO terms were then removed based on semantic similarity using rrvgo[118]. enrichR was used for gene symbol-based PANTHER pathway enrichment testing[119–122]. regioneR was utilized to perform permutation-based genomic coordinate enrichment testing through a randomized region strategy with 10,000 permutations[123]. GAT was used to perform random sampling-based genomic coordinate enrichment testing through 10,000 samplings of background regions[124]. Analysis of Motif Enrichment in the MEME Suite was utilized to perform transcription factor motif testing relative to background regions using the Human Methylcytosine database with rheMac10 sequences through the memes package[49,125–127]. The WGCNA package was used to construct a signed co-methylation network through the biweight mid-correlation (bicor) method[128,129].

## Reporting summary

Further information on research design is available in the Nature Research Reporting Summary linked to this article.

## Data availability

The sequencing data generated in this study have been deposited in the GEO database under accession code GSE171064. The metabolomics data generated in this study have been deposited in Dryad (https://doi.org/10.5061/dryad.c2fqz61bw). The lipidomics data generated in this study have been deposited in Dryad (https://doi.org/10.25338/B8C35T). Source data are provided with this paper.

## Code availability

All original code has been deposited on GitHub (https://github.com/ben-laufer/cffDNA-and-Brain-Manuscript) and Zenodo (https://doi.org/10.5281/zenodo.5348009).

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

## Acknowledgements

This work was supported by a National Institutes of Health (NIH) grant from the Eunice Kennedy Shriver National Institute of Child Health and Human Development [5R01HD084203-04] to C.K.W. and C.A.V., an NIH grant [R24OD010962] to JPC, a Canadian Institutes of Health Research (CIHR) postdoctoral fellowship [MFE-146824] and a CIHR Banting postdoctoral fellowship [BPF-162684] to BIL, the UC Davis Intellectual and Developmental Disabilities Research Center (IDDRC) [P50HD103526], and the California National Primate Research Center (CNPRC) [P51-OD011107]. This work was also supported by the USDA National Institute of Food and Agriculture Hatch Project 1021411, and the John E. Kinsella Chair in Food, Nutrition and Health (to C.M.S.). The Bruker Advance 600-MHz nuclear magnetic resonance Spectrometer is supported by NIH grant RR011973. The cffDNA research was funded by seed grants from the UC Davis IDDRC and Genome Center awarded to C.K.W. and BIL. The brain DNA library preparation was carried out by the DNA Technologies and Expression Analysis Cores at the UC Davis Genome Center and was supported by an NIH Shared Instrumentation Grant [1S10OD010786-01]. The authors would like to thank Jasmin Zarrabi and Dana Hill from the California National Primate Research Center for coordinating the project and we thank Dr. Blythe Durbin-Johnson for statistical guidance. The authors would also like to thank Dr. Brad Friedman from Genentech for invaluable discussions about the bioinformatic approaches presented in this manuscript.

## Author contributions

C.K.W., C.A.V., J.M.L., M.S.G., J.P.C., C.M.S., A.Y.T., J.A.V., and M.D.B. conceptualized the study. C.K.W. and C.A.V. obtained funding for the study. C.A.V. supervised all prenatal animal work and performed infant tissue sampling, and C.K.W. supervised and performed maternal gestation sampling. BIL prepared the pilot cffDNA libraries. B.I.L. and

H.H. performed the DNA methylation analyses under the supervision of J.M.L. Y.H. performed the metabolomic analyses under the supervision of C.M.S. Z.Z. performed the lipidomic analyses under the supervision of A.Y.T. L.A.D. performed the social stimuli behavior work under the supervision of J.P.C. C.E.H. performed the abstract stimuli behavior work under the supervision of M.D.B. and M.S.G. L.H. performed the immunology analyses under the supervision of J.A.V. B.I.L. wrote the manuscript with intellectual contributions from J.M.L. All authors wrote their respective methods section and reviewed and approved the final manuscript.

## Competing interests

The authors declare no competing interests.

## Additional information

[1]Department of Medical Microbiology and Immunology, School of Medicine, University of California Davis, Davis, CA 95616, USA. [2]UC Davis Genome Center, University of California, Davis, CA 95616, USA. [3]MIND Institute, School of Medicine, University of California Davis, Sacramento, CA 95817, USA. [4]Department of Food Science and Technology, University of California Davis, Davis, CA 95616, USA. [5]California National Primate Research Center, University of California Davis, Davis, CA 95616, USA. [6]Department of Psychiatry and Behavioral Sciences, School of Medicine, University of California Davis, Davis, CA 95616, USA. [7]Perinatal Origins of Disparities Center, University of California Davis, Davis, CA 95616, USA. [8]Department of Internal Medicine, University of California Davis, Davis, CA 95616, USA. [9]Department of Nutrition, University of California Davis, Davis, CA 95616, USA. [10]Department of Psychology, University of California Davis, Davis, CA 95616, USA. [11]Department of Obstetrics and Gynecology, School of Medicine, University of California Davis, Davis, CA 95616, USA. [12]Present address: Department of OMNI Bioinformatics, Genentech, Inc., South San Francisco, CA 94080, USA. [13]These authors contributed equally: Yu Hasegawa, Zhichao Zhang. ✉e-mail: jmlasalle@ucdavis.edu

