## [Peer Review File · Nature Communications]

Multi-omic brain and behavioral correlates of cell-free fetal DNA methylation in macaque maternal obesity modelsReviewers' comments:

Reviewer #1 (Remarks to the Author):

The authors examined the effects of maternal obesity on offspring neurodevelopment. They find that cell free fetal DNA has signatures that align with the infant brain. The public health implication are potentially high as obesity during pregnancy is a major issue in the US. This is a very novel study design and has some elegant aspects. My major concern about this work is the sample size – the results are being presented from a sample size of only a handful of animals. It appears that analysis does not adjust for several key potential confounders such as maternal age. Thus, the findings at the moment could be influenced by other factors beyond obesity.

Major

1. The authors should use caution in stating that the cfDNA is derived from the placenta. The PCA plot in Supp Figure 1 is not sufficient for this claim. The authors should use data from the genes that are specific to the placenta to support their argument. It is agreed that the increase of fetal derived DNA in Supp Figure 1 B does support fetal origin of the DNA, but the argument that this is placenta derived needs to be stronger.
2. For the data represented in Figure 3, which aim to show the difference in methylation between obese, interventions and controls, have any statistics been used to support the stated observation of decline? The authors should perform a statistical test to highlight these differences.
3. It is hard to assess the phenotypes of the offspring behavior in terms of the effect of the maternal obesity or the interventions. Was there an association between obesity during pregnancy and the behavior measures of the offspring? Were the interventions successful in reducing these effects? The authors should add this information into the manuscript as it provides a phenotype that they aim to ground their multi-omics data to.
4. Did the authors adjust for factors such as maternal age in their statistical models? If so this should be stated and if not, this is an issue as the sample size is quite small and adjusting for factors that can influence methylation in addition to obesity is critical.

Reviewer #2 (Remarks to the Author):

This study by Laufer and colleagues uses a rhesus macaque model to investigate mechanisms of maternal obesity leading to behavioral phenotypes relevant to neurodevelopment disorders. In addition to behavioral and metabolic measurements, the authors also measure methylation in maternally-derived fetal DNA as well as brain tissue. Interestingly (and perhaps surprisingly), the DMRs in cfDNA overlap with those in the brains of the infants. The authors then provide evidence for specific genetic risk factors and gene regulatory networks that may underlie some of the observed phenotypes.

Together, the study design and the results are important for understanding the environmental impacts on brain development and behavior. The overlap of the cfDNA DMRs and brain DMRs is itself a notable finding that opens up numerous avenues for studying gene x environment impacts on brain development in both model systems and humans.

I have some suggestions that could improve the manuscript:

Fig 1b-d: is it surprising that the brain regions at 6 months have the greatest # of DMRs for each comparison. Is this brain-specific or developmentally specific? If the authors measured DMRs from blood at 6 months, how many would they expect to observe? This also makes me wonder about the

origin of the cfDNA? What fetal tissues are they known to derive from? On a minor point for these figures, I suggest the authors note the exact # of DMRs above each highlighted colored (red or blue) bar in the histogram.

Figure 3a: It seems that the intervention and control separate well in the identified block in the cfDNA but there is less distinction in the brain. Could the authors comment on this? Are the calculations of DMRs that focused on this region only based on obesity vs control rather than intervention vs control?

Figure 3b-c: It seems a bit circular that mir663 would correlate with obesity if the DMR was identified based on obesity versus control. To me, the more important finding would be whether other traits do or do not correlate with mir663. The authors do note these other correlations, but it was not clear to me whether one or more were more relevant to NDD. Also, I really don't understand why the authors focused specifically on mir-663 versus the novel protein coding genes. This was not described.

Figure 4: The application of WGCNA to the WGBS data is a fairly novel approach to mining it. It might be interesting for the authors to contrast the regions/genes in the yellow module to the blue/DUX4 module. The yellow module is also negatively associated with obesity (and positive with pravastatin) but few other traits are significantly correlated. 4d: the correlations with maternal blood data are less satisfying here. There is not a strong consistent theme that one could link to a developmental time line unless this could be made more clear. It seems almost like a grab bag of various correlations. The brain data are more compelling.

Reviewer #3 (Remarks to the Author):

Review of Manuscript # NCOMMS-21-34179

-This manuscript is interesting and provides important data towards understanding signaling at the maternal-fetal interface, which is incompletely understood. The authors state that DNA methylation profiles can distinguish placenta samples from newborns diagnosed later with ASD from controls. They also state that it has been shown that placental and embryonic brain DNA profiles are similar in a mouse model. They suggest that placental methylation can inform on NDD/ASD risk, however this type of sampling is invasive and comes with risk. This paper investigates the potential for non-invasive prenatal testing by using cfDNA that is representative of its placental origin.

-The cfDNA data is well described and comprehensive showing correlation with altered infant brain and behaviors.

-The metabolomics and lipidomics correlates are very interesting, but these data prove to be a weak part of the manuscript that needs to be addressed.

-The methylation data correlates with obesity with caloric restriction and pharmacological intervention mitigating the dysfunctional methylation. What about the metabolite and lipid data, are intermediate levels observed for these groups as compared to control and obese? It seems from the correlation plots in 4b this is true, but more clarity could be included.

-Why was pravastatin chosen for the pharmacological intervention? There is no explanation of reasoning for why this drug or its dose was chosen.

-What were the body weights of the groups? What was the caloric restriction and its effect on weight? What was the effect of the pravastatin treatment on weight?

-Linoleic acid, in addition to being an omega-6 polyunsaturated fatty acid, is also an essential fatty acid meaning it is derived exclusively from the diet. How does the diet/food intake of different groups affect the levels of this diet-derived fatty acid? This also applies to C20:3 homo-g-linolenic acid which

is a product of desaturation and then elongation of linoleic acid.

-Line 240-242 – says that a list of metabolites are correlated but none of them are significant. This should be reworded, and non-significant correlations should not be stated as correlations.

-Were any expression or functional assays done to validate the predicted changes in function due to the differentially methylated genes?

-Figure 3c – many of the lipids chosen for inclusion are not significantly correlated except the C20:3 Homo- γ -linolenic acid/8,11,14-eicosatrienoic acid in prefrontal cortex. Why were these chosen for the table – what was the criteria for inclusion? Same comment for metabolites, why are metabolites that are not significant for any region included in the table?

-The legend and/or text should be clarified that the correlation plots in 4b include all the different groups. It also might be nice to color code the correlation plot to show which points correspond to which group.

-The method description for the metabolomics and lipidomics is insufficient and should include the relevant detail used to carry out this series of experiments. The references provide some information but there is too much ambiguity about how the experiments were performed, which could hamper attempts to reproduce the data.

-Was the lipidomics assay validated for analytical performance? If so, please state the basic assay performance attributed that were characterized.

-There are numerous missing details that make it hard to evaluate the confidence of the lipid analysis and measurement. The lipidomics assay utilizes GC-FID, which is a non-specific detector. The methods section says the FAMES are identified according to their retention time using a mix of 29 FAME standards, but these standards are not defined and no example chromatogram is provided to show the separation and identification of the FAMES identified in the paper. No representative chromatogram of the data is shown to demonstrate the relative abundance and specificity (or not) of the data. It is not clear what the internal standard was and how/if it was used in the calculation of the concentration of FAMES. The actual concentration and the linear ranges, LOQ/LOQ for each FAME are not provided.

-Were any of the FAME identities confirmed via mass spectrometry? At best these GC-FID identifications are putative, and should be stated as such. If additional attributes were confirmed (e.g., accurate mass, fragmentation, match with an authentic standard assayed under the same conditions) then this should be provided and a definitive ID could be stated.

-The methods section for the lipidomics says the authors used “30 mg of homogenized half brain” were used but Fig 3c and 4a show specific regions of the brain. The methods should be amended to reflect the regions assayed and which half of the brain was assayed. Also, 30 mg is a small amount of tissue and it should be stated if the tissue was taken from the same region of the brain area similarly for all groups or no attention was paid to this variable.

-Was the metabolomics assay (NMR) validated for analytical performance? If so, please state the basic assay performance attributed that were characterized. Were any of these metabolites confirmed with authentic standards? What were the measurement ranges for each of the analytes, LOD/LOQ? Similar comments for this assay as for lipidomics, no representative data is shown for standards and representative data.

-What were the actual measured levels of the metabolites and lipids detected (as shown in Fig 3, 4, or supp 2,3,5,6)? There should be a supplemental table of the actual values quantified for each metabolite or lipid in either plasma or brain region(s) with the variability (e.g., mean value +/- standard deviation). If quantitation was not absolute (i.e., relative quantification) this should be stated and it should be stated what attribute was used for correlation (peak area, intensity, etc.)

-Abstract should not say “immune and metabolic biomarkers” unless the molecule has a defined relationship with a normal biologic process, a pathogenic process, or a biologic response to an exposure or intervention (PMID 27010052). The molecules described within the manuscript have correlations with various parameters, but are not necessarily biomarkers and the text should reflect this distinction.

Reviewer #4 (Remarks to the Author):

The authors present correlative data between differential DNA methylation patterns in cffDNA, which is claimed to be largely from placental sources, and methylation within area of the brains of male offsprings derived from pregnancies in lean, obese, obese-calorically restricted and pravastatin treated mothers. After birth, it appears that the offspring with nursed by foster mothers. The DMR are claimed to be significantly overlapping with DMR in the brains of male offspring. They provide some evidence for an association of DMRs from a large complex containing mir-663 in cffDNA and brain methylation along with a DUX4 containing region with maternal obesity and offspring behavior. This reviewer found the presentation to be confusing. This was contributed by the absence of labeling of the figures and especially the tables in the submission. While it is clear why, the number of animals in each group were quite small. Given the small ‘n’, large ‘p’, and it was difficult to understand how the authors gained confidence in the identified DMRs

Specific comments.

1. What are the conditions of the foster mothers? Could the fostering have an influence on the offspring?
2. What about the fathers? What were their weight/obesity status?
3. Supplemental Figure 1 is important but does not convincingly prove that the cffDNA is ‘closer’ to placenta than cfDNA. How is this statement supported? The individual points for each animal wander all over the PCA plot and one would assume that there would be some consistency over time, that is a convergence at later gestational time points as the placental mass rises. In addition, why wasn’t cfDNA plotted at each gestational time point?
4. Similarly, despite the statement ‘The overlaps are not only apparent across different pregnancy timepoints and brain regions for their respective sources, but a subset also converge between cffDNA and brain.’, the data in Figure 1b-1d does not appear to show consistent association in the methylation profiles from cffDNA across pregnancy time points nor are consistent relationships between cffDNA with the different brain regions. How do the authors define ‘consistent’ in this context? On the other hand, the brain regions seem to be more consistently related, which would be expected. How was multiple testing addressed in this context, given that the ‘n’ samples were quite small.
5. Further if one questions the security of these relationships, one must also question the step of [merging] the genomic coordinates of the DMRs across all pairwise comparisons (were merged) into separate consensus regions for cffDNA and brain and the same was done for their respective background regions. If you take all the ‘significant’ DMR regions, compile them and then recompare them, it is not surprising that one might find association after permutation analysis as even random associations are now ‘enriched’ even though the individuals DMRs could be random. This association then seemingly could be propagated to the GO ontology associations with brain and cffDNA as they were ‘predestined’ to contain the same information. Thus, using GO overlap to support that the reliability of the cffDNA DMR compared to brain DRM pathways seems erroneous.
6. It is unclear how the large block overlapping mir-663 was accomplished as a ‘separate analysis’. In the analysis referenced (ref 30), the DMR required 10% methylation difference between samples in at least three CpGs within 300 base pairs and a P-value of <0.05. What were the criteria in the identification of this large block containing mir-663? How many methylation sites were there and was there consistency in differential methylation in this large block?
7. The authors contentions about the relationships between DMR and genes would be strongly supported by assessment of mRNA levels. Were these analyses performed?

We thank the four expert reviewers for their time and helpful suggestions that have led to an improved manuscript. We note that all four reviewers commented on the significance and novelty of our study. This was apparent in comments such as “public health implication are potentially high...This is a very novel study design and has some elegant aspects” (Rev 1), “the study design and the results are important ... overlap of the cffDNA DMRs and brain DMRs is itself a notable finding that opens up numerous avenues for studying gene x environment impacts on brain development in both model systems and humans” (Rev 2), “...important data towards understanding signaling at the maternal-fetal interface, which is incompletely understood...cffDNA data is well described and comprehensive showing correlation with altered infant brain and behaviors” (Rev 3), and “...evidence for an association of DMRs from a large complex containing mir-663 in cffDNA and brain methylation along with a *DUX4* containing region with maternal obesity and offspring behavior” (Rev 4).

In order to address the constructive critiques from the reviewers, we have modified 3 of the main figures, which included adding a new panel that details a third statistical analysis of the novel *mir-663* block. We have also added a new panel to a supplementary figure to add confidence to the placental origin of the cffDNA along with revisions to the text to highlight that this point has been shown in the past by multiple publications. We have also added a new supplementary figure with the FAME standards along with 3 new supplementary tables that contain data and analyses related to maternal traits, the multi-omics measurements, and the behavior analyses. Finally, we have extensively revised the text to incorporate the above modifications and further clarify key details related to the methods and related statistical analyses. Please see the detailed point-by-point response below along with the attached revised manuscript with changes shown in red.

Reviewer #1 (Remarks to the Author):

The authors examined the effects of maternal obesity on offspring neurodevelopment. They find that cell free fetal DNA has signatures that align with the infant brain. The public health implication are potentially high as obesity during pregnancy is a major issue in the US. This is a very novel study design and has some elegant aspects. My major concern about this work is the sample size – the results are being presented from a sample size of only a handful of animals. It appears that analysis does not adjust for several key potential confounders such as maternal age. Thus, the findings at the moment could be influenced by other factors beyond obesity.

Response: It is not possible to increase the sample size of our study because of the challenges with introducing new variables with a new cohort, as well as the cost. In the revised Discussion, we now better justify the sample size by a comparison to other developmental nonhuman primate (NHP) studies summarized in a recent review (1). Our study is controlled for sex (males only) and uses double the sample size of prior epigenomic studies performed in perinatal NHP models (5-7 per group vs 3 per group). Our study is a typical sample size for those performing phenotypic outcomes (3-8 per group) according to this review. It is also important to emphasize the unique insights that this NHP study has provided to the field that was not possible in either prior human or rodent studies in which larger samples sizes were more feasible. Namely, the ability to longitudinally compare methylation patterns in cell free fetal DNA with those of postnatal infant brain, a feat not possible in human studies. In addition, while our experimental design shares advantages with rodent maternal obesity models in the ability to control for diet and housing, we have identified two genomic loci in this study (*mir-663* and *DUX4*) that have primate-specific genomic organization and regulation that would have not been detectable in a rodent model. We address the concern about potential confounders in the response to point #4 below.

Major

1. The authors should use caution in stating that the cffDNA is derived from the placenta. The PCA plot in Supp Figure 1 is not sufficient for this claim. The authors should use data from the genes that are specific to the placenta to support their argument. It is agreed that the increase of fetal derived DNA in Supp Figure 1 B does support fetal origin of the DNA, but the argument that this is placenta derived needs to be stronger.

Response: We would like to clarify that the claim of cffDNA being of placenta origin was not only based on our own data. It has been extensively shown in the literature that cffDNA is derived from placental trophoblast cells, a finding based on both genetic and epigenetic evidence (2-7). While we had cited these papers in the original manuscript, we have expanded the discussion of the evidence supporting the placental origin of cffDNA in the revised Introduction: “Genetic evidence from cases of anembryonic pregnancies or confined placental mosaicism have demonstrated that cffDNA originates from the trophoblasts of the placenta.^{35–37} Epigenetic evidence has confirmed the placental origin of cffDNA through the detection of hypomethylated domains called partially methylated domains, which are characteristic of placenta.^{30,41}”

In addition, we have included a new panel “c” to Supplementary Fig. 1 that shows a global analysis of WGBS data from the main experiment, demonstrating the significant hypomethylation of cffDNA compared to brain, consistent with the placental origin of cffDNA. We have added this sentence to the Results: “Lastly, using a 20 kb windowing approach to assess global methylation distributions, we compared cffDNA, cfDNA and brain WGBS datasets from the main experiment (Fig. 1a), and showed that cffDNA was hypomethylated compared to brain (Supplementary Fig. 1c), consistent with placental origin.”

2. For the data represented in Figure 3, which aim to show the difference in methylation between obese, interventions and controls, have any statistics been used to support the stated observation of decline? The authors should perform a statistical test to highlight these differences.

Response: The specific region was identified based on pairwise block comparisons and the statistics for those are presented in Supplementary Table 9, where we used a significance threshold of empirical $p < 0.05$. Panels b and c of Fig. 3 contain correlation analyses with correlation coefficients (r) and p -values, and the colors show direction of difference for each group. There were significant ($p < 0.05$) correlations for the maternal obesity groups. However, since multiple reviewers requested clarifications about the evidence plotted in Figure 3, we have also analyzed the results by a third statistical linear mixed effects model that included maternal group, cffDNA timepoint or brain region, maternal age, cohort, birth gestational day, delivery mode, and foster status as fixed effects and individual as a random effect using the smoothed methylation average across all the CpG sites within the main region of the block. These data are now shown in the new panel 3d and those results demonstrate a trend ($p = 0.07$) for a difference between Obese vs Control in cffDNA and a significant ($p = 0.01$) difference between Obese vs Caloric Restriction in the brain.

3. It is hard to assess the phenotypes of the offspring behavior in terms of the effect of the maternal obesity or the interventions. Was there an association between obesity during pregnancy and the behavior measures of the offspring? Were the interventions successful in reducing these effects? The authors should add this information into the manuscript as it provides a phenotype that they aim to ground their multi-omics data to.

Response: We now include the raw results of all measurements in Supplementary Table 2. For the recognition memory test of abstract stimuli, there was a significant group effect ($p = 0.04$) by ANOVA as well as a significant difference between obese and lean control groups by post hoc pairwise Tukey HSD. For the recognition memory test of social stimuli, there was a nonsignificant trend toward a group difference ($p=0.2$), with infants from untreated obese dams showing an increased memory of social novelty compared to those from lean dams and both treatment groups. Typically, a larger number of animals is needed for statistical significance in behavioral testing compared to molecular genomic analyses. We would argue that the marginally significant behavioral findings actually improve the impact of our DNA methylation findings and highlight the utility of our correlation analyses because there are known differences in neurodevelopmental outcomes between human infants born to mothers with obesity. Specifically, not every child born to a mother with obesity develops a neurodevelopmental disorder. But the approach of using a gene-specific epigenetic difference measured during pregnancy may be a more sensitive indicator of future behavioral outcomes than just using maternal pre-pregnancy BMI or weight as a risk factor. In a revised manuscript, we have expanded upon this concept in the Discussion with a new paragraph discussing limitations and strengths of the study.

4. Did the authors adjust for factors such as maternal age in their statistical models? If so this should be stated and if not, this is an issue as the sample size is quite small and adjusting for factors that can influence methylation in addition to obesity is critical.

Response: The new Supplementary Tables 1 and 2 contain additional information about the animals in the study, including maternal age. The groups were matched for maternal age and other variables as much as was possible and all offspring were male. While it was not possible to adjust for all of the variables in the pairwise methylome analyses due to the limitations of utilizing complex statistical models with current DMR/block detection approaches, the addition of the results of a linear mixed effects model containing these covariates in Figure 3d demonstrate that the main results in Figure 3 have not been overly influenced factors other than maternal obesity. We have now added a statement about these findings to the results section.

Reviewer #2 (Remarks to the Author):

This study by Laufer and colleagues uses a rhesus macaque model to investigate mechanisms of maternal obesity leading to behavioral phenotypes relevant to neurodevelopment disorders. In addition to behavioral and metabolic measurements, the authors also measure methylation in maternally-derived fetal DNA as well as brain tissue. Interestingly (and perhaps surprisingly), the DMRs in cffDNA overlap with those in the brains of the infants. The authors then provide evidence for specific genetic risk factors and gene regulatory networks that may underlie some of the observed phenotypes.

Together, the study design and the results are important for understanding the environmental impacts on brain development and behavior. The overlap of the cffDNA DMRs and brain DMRs is itself a notable finding that opens up numerous avenues for studying gene x environment impacts on brain development in both model systems and humans.

I have some suggestions that could improve the manuscript:

Fig 1b-d: is it surprising that the brain regions at 6 months have the greatest # of DMRs for each comparison. Is this brain-specific or developmentally specific? If the authors measured DMRs from blood at 6 months, how many would they expect to observe? This also makes me wonder

about the origin of the cffDNA? What fetal tissues are they known to derive from? On a minor point for these figures, I suggest the authors note the exact # of DMRs above each highlighted colored (red or blue) bar in the histogram.

Response: The placental origin of cffDNA has been extensively shown in the literature, a conclusion based on both genetic and epigenetic evidence. While we had cited these papers in the original manuscript, we have expanded the discussion of the evidence supporting the placental origin of cffDNA in the revised Introduction: “Genetic evidence from cases of anembryonic pregnancies or confined placental mosaicism have demonstrated that cffDNA originates from the trophoblasts of the placenta.^{35–37} Epigenetic evidence has confirmed the placental origin of cffDNA through the detection of hypomethylated domains called partially methylated domains, which are characteristic of placenta.^{30,41}”

In addition, we have included a new panel “c” to Supplementary Figure 1 that shows a global analysis of WGBS data from the main experiment, demonstrating hypomethylation of cffDNA compared to brain and cfDNA, consistent with the placental origin of cffDNA. We have added this sentence to the Results: “Lastly, using a 20 kb windowing approach to assess global methylation distributions, we compared cffDNA, cfDNA, and brain WGBS datasets from the main experiment (Fig 1a), and showed that cffDNA was significantly hypomethylated compared to cfDNA and brain (Supplementary Fig. 1c), consistent with placental origin.”

For the question of why there are more DMRs identified in brain than cffDNA, this is likely related to technical rather than biological differences. We applied a 10% methylation difference cut-off to the cffDNA DMRs compared to 5% for brain DMRs, to be consistent with the DMR results from human brain and placenta samples in our previous studies. These details about the cut-offs are currently mentioned in the Methods. Finally, we have also added the number of DMRs to the highlighted colors in the histogram.

Figure 3a: It seems that the intervention and control separate well in the identified block in the cffDNA but there is less distinction in the brain. Could the authors comment on this? Are the calculations of DMRs that focused on this region only based on obesity vs control rather than intervention vs control?

Response: The specific region plotted in Figure 3a was identified based on pairwise comparisons of all groups and the statistics for those are presented in Supplementary Table 9. Additionally, the correlation analyses in 3c demonstrate the distinction between Obesity and Caloric Restriction in the brain. To further emphasize these group differences, we have added a third statistical analysis and provide the data in the new Figure 3d and detail the results in the revised manuscript.

Figure 3b-c: It seems a bit circular that mir663 would correlate with obesity if the DMR was identified based on obesity versus control. To me, the more important finding would be whether other traits do or do not correlate with mir663. The authors do note these other correlations, but it was not clear to me whether one or more were more relevant to NDD. Also, I really don't understand why the authors focused specifically on mir-663 versus the novel protein coding genes. This was not described.

Response: We have expanded the rationale for the behavioral tasks to NDD/ASD in the beginning of the Results section and again when describing the significant correlations of miR-663 block methylation in the Discussion. Our focus on mir-663 was simply because this is the best characterized transcript in the locus from human studies. In the revised manuscript, we added a justification within the Results and mention this again in the Discussion.

Figure 4: The application of WGCNA to the WGBS data is a fairly novel approach to mining it. It might be interesting for the authors to contrast the regions/genes in the yellow module to the blue/DUX4 module. The yellow module is also negatively associated with obesity (and positive with pravastatin) but few other traits are significantly correlated. 4d: the correlations with maternal blood data are less satisfying here. There is not a strong consistent theme that one could link to a developmental time line unless this could be made more clear. It seems almost like a grab bag of various correlations. The brain data are more compelling.

Response: We have added Supplementary Table 11, which contains the regions from the annotated regions from the yellow module. We agree that the maternal blood data highlights a complex temporal pattern without a consistent marker across pregnancy; however, we believe that this is a noteworthy observation within itself. We have also revised Fig. 4 a & d to focus on the most significant correlations.

Reviewer #3 (Remarks to the Author):

Review of Manuscript # NCOMMS-21-34179

-This manuscript is interesting and provides important data towards understanding signaling at the maternal-fetal interface, which is incompletely understood. The authors state that DNA methylation profiles can distinguish placenta samples from newborns diagnosed later with ASD from controls. They also state that it has been shown that placental and embryonic brain DNA profiles are similar in a mouse model. They suggest that placental methylation can inform on NDD/ASD risk, however this type of sampling is invasive and comes with risk. This paper investigates the potential for non-invasive prenatal testing by using cfDNA that is representative of its placental origin.

-The cfDNA data is well described and comprehensive showing correlation with altered infant brain and behaviors.

-The metabolomics and lipidomics correlates are very interesting, but these data prove to be a weak part of the manuscript that needs to be addressed.

-The methylation data correlates with obesity with caloric restriction and pharmacological intervention mitigating the dysfunctional methylation. What about the metabolite and lipid data, are intermediate levels observed for these groups as compared to control and obese? It seems from the correlation plots in 4b this is true, but more clarity could be included.

Response: The levels of metabolite and lipids is now included in Supplementary Tables 2-3. Each metabolite is different. Some metabolites show intermediate levels in the treatment groups, but not all. The metabolite data will be described in greater detail in two other manuscripts, including a manuscript currently in revision that shows significant differences between lean and obese dams in metabolites involved in one carbon metabolism and energy metabolism.

-Why was pravastatin chosen for the pharmacological intervention? There is no explanation of reasoning for why this drug or its dose was chosen.

Response: We added this sentence to the revised Introduction "Obese women have a higher risk of developing gestational hypertension and preeclampsia.", as well as this sentence to the first paragraph of the Results to provide justification "Pravastatin is clinically used to lower blood

pressure, and recently, it has received attention as it may attenuate gestational hypertension and the risk of developing preeclampsia.”

-What were the body weights of the groups? What was the caloric restriction and its effect on weight? What was the effect of the pravastatin treatment on weight?

Response: A new Supplementary Table 1 is included, containing additional information about the animals in the study, including body weights and statistical comparisons. We have also added the following sentences to the beginning of the Results section that describe the results: “Compared to obese control and both obese treatment groups, the lean control group showed significantly ($p < 0.05$) pre-pregnancy reduced body composition score (BCS) and significantly reduced body weights at all four gestational timepoints (Supplementary Table 1).”

-Linoleic acid, in addition to being an omega-6 polyunsaturated fatty acid, is also an essential fatty acid meaning it is derived exclusively from the diet. How does the diet/food intake of different groups affect the levels of this diet-derived fatty acid? This also applies to C20:3 homo-g-linolenic acid which is a product of desaturation and then elongation of linoleic acid.

Response: There were no statistically significant differences across the four groups for all measured fatty acid concentrations. Dietary intake also did not affect offspring brain fatty acid concentration.

-Line 240-242 – says that a list of metabolites are correlated but none of them are significant. This should be reworded, and non-significant correlations should not be stated as correlations.

Response: Asparagine and citrate were significant ($p < 0.05$) and the others were notable trends ($p < 0.08$). The plots present the correlation coefficient (r) above the p -value, which is in parenthesis. We revised the text to focus on significant findings as correlations.

-Were any expression or functional assays done to validate the predicted changes in function due to the differentially methylated genes?

Response: We have not performed expression or functional follow-up to the loci discovered from an unbiased methylome analysis. While important, these analyses are outside the scope of the currently funded study which was to use DNA methylation as the best interface to reflect genetic by *in utero* environment. In our past human studies comparing transcriptomes to methylomes of the same samples from *in utero* exposures, as well as those of published studies of others, methylomes provide more information about past exposures and early developmental processes (9-12). In contrast, gene expression is only captured at the time of tissue collection and may not reflect differences relating to gene expression in response to specific environmental experiences.

-Figure 3c – many of the lipids chosen for inclusion are not significantly correlated except the C20:3 Homo- γ -linolenic acid/8,11,14-eicosatrienoic acid in prefrontal cortex. Why were these chosen for the table – what was the criteria for inclusion? Same comment for metabolites, why are metabolites that are not significant for any region included in the table?

Response: For the lipids, we chose to focus on all fatty acids that were either abundant in brain or the intermediates of long chain PUFA synthesis that are important in brain development. The metabolites were more numerous, so for inclusion in Fig 3c, we filtered for $p < 0.08$ and a high absolute correlation coefficient. In the revised figure, we have removed the lipids and

metabolites that did not pass the $p < 0.05$ threshold. We have also included this rationale for inclusion in the revised figure legend.

-The legend and/or text should be clarified that the correlation plots in 4b include all the different groups. It also might be nice to color code the correlation plot to show which points correspond to which group.

Response: We revised the figure legend for 4b to clarify this point.

-The method description for the metabolomics and lipidomics is insufficient and should include the relevant detail used to carry out this series of experiments. The references provide some information but there is too much ambiguity about how the experiments were performed, which could hamper attempts to reproduce the data.

Response: For the metabolomics and lipidomics, we expanded the description in the Methods and provide new Supplementary Tables (2-3) that contain the concentrations.

-Was the lipidomics assay validated for analytical performance? If so, please state the basic assay performance attributed that were characterized.

Response: The fatty acid analysis performed in this study has been validated in previous rodent brain and food meat studies (13, 14). Example chromatograms of separations of FAMES using this method has been provided previously (14).

-There are numerous missing details that make it hard to evaluate the confidence of the lipid analysis and measurement. The lipidomics assay utilizes GC-FID, which is a non-specific detector. The methods section says the FAMES are identified according to their retention time using a mix of 29 FAME standards, but these standards are not defined and no example chromatogram is provided to show the separation and identification of the FAMES identified in the paper. No representative chromatogram of the data is shown to demonstrate the relative abundance and specificity (or not) of the data. It is not clear what the internal standard was and how/if it was used in the calculation of the concentration of FAMES. The actual concentration and the linear ranges, LOQ/LOQ for each FAME are not provided.

Response: We now list the 29 FAME standards in the methods and provide a chromatogram (Supplementary Figure 7). C 17:0 PC internal standard was used for quantification as described in the manuscript. GC-FID has shown to accurately identify and quantify fatty acids as described in the literature (15). In the revised manuscript, we have expanded the Methods to better describe GC-FID.

-Were any of the FAME identities confirmed via mass spectrometry? At best these GC-FID identifications are putative, and should be stated as such. If additional attributes were confirmed (e.g., accurate mass, fragmentation, match with an authentic standard assayed under the same conditions) then this should be provided and a definitive ID could be stated.

Response: In the revised manuscript, we have expanded the Methods to describe how we adapted this acid catalyst FAME GC-FID method. Specifically, the LOQ is at least 10 times the signal to noise ratio for each identified FAME and the equation $\text{mass of FA/mass of internal standard C17:0 PC} = \text{area of FA/area of C17:0}$ was used to calculate each FA concentration. The comparative study of GC-FID vs GC-MS has been studied and yielded similar result (16).

-The methods section for the lipidomics says the authors used "30 mg of homogenized half brain" were used but Fig 3c and 4a show specific regions of the brain. The methods should be amended to reflect the regions assayed and which half of the brain was assayed. Also, 30 mg is a small amount of tissue and it should be stated if the tissue was taken from the same region of the brain area similarly for all groups or no attention was paid to this variable.

Response: In the revised Methods lipidomics section, we edited the text to reflect the sampling of the three brain regions (hippocampus, hypothalamus, prefrontal cortex) from matched regions. We also describe an optimized extraction protocol for lipid and polar metabolites that has been developed by our group using pig brain (17).

-Was the metabolomics assay (NMR) validated for analytical performance? If so, please state the basic assay performance attributed that were characterized. Were any of these metabolites confirmed with authentic standards? What were the measurement ranges for each of the analytes, LOD/LOQ? Similar comments for this assay as for lipidomics, no representative data is shown for standards and representative data.

Response: In the revised Methods section on metabolomics, we describe how the NMR metabolomics methods have been previously validated and published (18-20). NMR-based metabolomics has shown high repeatability and reproducibility with low CV's of technical replicates, as well as low %error in reported concentrations (20). The concentrations of individual metabolites are determined based on the addition of an internal standard as described in the methods. LOD / LOQ vary depending on the metabolite and sample matrix, but is generally between 1 μ M and 50 μ M, with most metabolites being able to reliably measured between 1 and 10 μ M (18).

-What were the actual measured levels of the metabolites and lipids detected (as shown in Fig 3, 4, or supp 2,3,5,6)? There should be a supplemental table of the actual values quantified for each metabolite or lipid in either plasma or brain region(s) with the variability (e.g., mean value +/- standard deviation). If quantitation was not absolute (i.e., relative quantification) this should be stated and it should be stated what attribute was used for correlation (peak area, intensity, etc.)

Response: We have added the concentrations to the new Supplementary Tables 2-3.

-Abstract should not say "immune and metabolic biomarkers" unless the molecule has a defined relationship with a normal biologic process, a pathogenic process, or a biologic response to an exposure or intervention (PMID 27010052). The molecules described within the manuscript have correlations with various parameters, but are not necessarily biomarkers and the text should reflect this distinction.

Response: We reworded the abstract and manuscript text to avoid the use of the word "biomarker" when referring to the DNA methylation results.

Reviewer #4 (Remarks to the Author):

The authors present correlative data between differential DNA methylation patterns in cffDNA, which is claimed to be largely from placental sources, and methylation within area of the brains of male offsprings derived from pregnancies in lean, obese, obese-calorically restricted and

pravastatin treated mothers. After birth, it appears that the offspring with nursed by foster mothers. The DMR are claimed to be significantly overlapping with DMR in the brains of male offspring. They provide some evidence for an association of DMRs from a large complex containing mir-663 in cffDNA and brain methylation along with a DUX4 containing region with maternal obesity and offspring behavior.

This reviewer found the presentation to be confusing. This was contributed by the absence of labeling of the figures and especially the tables in the submission. While it is clear why, the number of animals in each group were quite small. Given the small 'n', large 'p', and it was difficult to understand how the authors gained confidence in the identified DMRs

Specific comments.

1. What are the conditions of the foster mothers? Could the fostering have an influence on the offspring?

Response: The new Supplementary Tables 1-3 contain additional information about the animals in the study, including fostering. We have added a third statistical approach and provide the new Figure 3d, which contains the results of pairwise comparisons from a linear mixed effects model that includes maternal group, cffDNA timepoint or brain region, maternal age, cohort, birth gestational day, delivery mode, and foster status as fixed effects and individual as a random effect. This analysis demonstrates that the results have not been overly influenced by factors other than maternal obesity, despite current DMR/block calling methods not being able to support this type of static modeling.

2. What about the fathers? What were their weight/obesity status?

Response: The new Supplementary Table 1 contain additional information about the animals in the study, including the body compositions scores of the sires.

3. Supplemental Figure 1 is important but does not convincingly prove that the cffDNA is 'closer' to placenta than cfDNA. How is this statement supported? The individual points for each animal wander all over the PCA plot and one would assume that there would be some consistency over time, that is a convergence at later gestational time points as the placental mass rises. In addition, why wasn't cfDNA plotted at each gestational time point?

Response: The placental origin of cffDNA is not a novel finding or a biological question of this manuscript. It has been extensively shown in the literature that cffDNA is derived from placental trophoblast cells, a finding based on both genetic and epigenetic evidence (2-7). While we had cited these papers in the original manuscript, we have expanded the discussion of the evidence supporting the placental origin of cffDNA in the revised Introduction: "Genetic evidence from cases of anembryonic pregnancies or confined placental mosaicism have demonstrated that cffDNA originates from the trophoblasts of the placenta.³⁵⁻³⁷ Epigenetic evidence has confirmed the placental origin of cffDNA through the detection of hypomethylated domains called partially methylated domains, which are characteristic of placenta.^{30,41}"

In addition, we have included a new panel "c" to Supplementary Fig. 1 that shows a global analysis of WGBS data from the main experiment, demonstrating the hypomethylation of cffDNA compared to cfDNA brain, consistent with the placental origin of cffDNA. We have added this sentence to the Results: "Lastly, using a 20 kb window approach to assess global methylation distributions, we compared cffDNA, cfDNA, and brain WGBS datasets from the main experiment, and showed that cffDNA was hypomethylated compared to both and cfDNA and brain (Supplementary Fig. 1c)."

We have revised the legend for Supplementary Fig. 1a to emphasize that it was from a pilot experiment of lower coverage WGBS designed to test feasibility, and that the cfDNA was

derived from a non-pregnant female using the same method. Therefore, cfDNA cannot be measured across gestation because it was from a nonpregnant female. Finally, the Trimester 1 time point was not available when this initial technical pilot focused on library preparation was conducted.

4. Similarly, despite the statement 'The overlaps are not only apparent across different pregnancy timepoints and brain regions for their respective sources, but a subset also converge between cffDNA and brain.', the data in Figure 1b-1d does not appear to show consistent association in the methylation profiles from cffDNA across pregnancy time points nor are consistent relationships between cffDNA with the different brain regions. How do the authors define 'consistent' in this context? On the other hand, the brain regions seem to be more consistently related, which would be expected. How was multiple testing addressed in this context, given that the 'n' samples were quite small.

Response: We agree that the precise overlap of individual DMRs between cffDNA and brain are not consistent across all gestational time points, but we do not attempt to make this claim in the manuscript. Our claim is that some cffDNA-brain overlap is observed across different gestational ages and that these regions converge on key functions, pathways, and transcription factors. We note that the overlap of the consensus cffDNA and brain DMRs is statistically significant using two different permutation-based approaches that do not involve multiple testing. For the DMR calling, multiple testing is controlled for through a permutation-based approach that provides empirical p -values. We have further expanded on these points in the revised manuscript.

5. Further if one questions the security of these relationships, one must also question the step of [merging] the genomic coordinates of the DMRs across all pairwise comparisons (were merged) into separate consensus regions for cffDNA and brain and the same was done for their respective background regions. If you take all the 'significant' DMR regions, compile them and then recompare them, it is not surprising that one might find association after permutation analysis as even random associations are now 'enriched' even though the individuals DMRs could be random. This association then seemingly could be propagated to the GO ontology associations with brain and cffDNA as they were 'predestined' to contain the same information. Thus, using GO overlap to support that the reliability of the cffDNA DMR compared to brain DRM pathways seems erroneous.

Response: As the reviewer mentions, in addition to passing a standard false discovery rate ($q < 0.05$), the GO analysis also presents a more stringent filtering criteria, which was a FWER threshold based on 100 random samplings of the background regions. These two approaches address false discovery distinctly and give the same results. Additionally, we have performed each GO analysis separately and obtained similar results but for the main figures, we sought to simplify the results, given the multiple pairwise contrasts. Finally, the merging of regions from multiple contrasts was determined to be more appropriate than a meta-analysis, which we also performed and gave similar, although inflated results. Thus, the main finding of GO term overlaps between cffDNA and brain is consistently replicated across different approaches, including those that do not involve merging regions.

We would also like to emphasize that each set of DMRs and background regions was different for the cffDNA and brain, and thus they represent a different subset of the genome. If the results were random and biased, then we would not expect the same result across different sample sources. We explain this important aspect of the approach in greater detail in the revised manuscript: "Next, the genomic coordinates of the DMRs, which were determined independently from each pairwise comparison, were merged into separate consensus regions

for cffDNA and brain and the same was done for their respective background regions, also determined independently in each comparison.”

6. It is unclear how the large block overlapping mir-663 was accomplished as a ‘separate analysis’. In the analysis referenced (ref 30), the DMR required 10% methylation difference between samples in at least three CpGs within 300 base pairs and a P-value of <0.05. What were the criteria in the identification of this large block containing mir-663? How many methylation sites were there and was there consistency in differential methylation in this large block?

Response: The separate block analyses offer a “zoomed out” perspective through a modified region-level model that differs from the DMR calling analyses, and the full results containing the number of CpGs and related metrics are presented in Supplementary Table 9. There are a number of different arguments provided to the block approach, which are detailed in the publicly available code and in the dmrseq vignette. We have added additional information about the approach to the methods and added the new Figure 3d that represents a third statistical analysis of the region.

7. The authors contentions about the relationships between DMR and genes would be strongly supported by assessment of mRNA levels. Were these analyses performed?

Response: We have not performed expression or functional follow-up to the loci discovered from an unbiased methylome analysis. While important, these analyses are outside the scope of the currently funded study which was to use DNA methylation as the best interface to reflect genetic by *in utero* environment. In our past human studies comparing transcriptomes to methylomes of the same samples from *in utero* exposures, as well as those of published studies of others, methylomes provide more information about past exposures and early developmental processes (8-11). In contrast, gene expression is only captured at the time of tissue collection and may not reflect differences relating to gene expression in response to specific environmental experiences. Because this was a multi-omics analysis that integrated immune, metabolome, and lipidome of the same limited samples, we prioritized the limited remaining tissue for the more informative DNA methylome rather than transcriptome analysis.

References:

1. H. F. Huber, S. L. Jenkins, C. Li, P. W. Nathanielsz, Strength of nonhuman primate studies of developmental programming: review of sample sizes, challenges, and steps for future work. *J Dev Orig Health Dis* **11**, 297-306 (2020).
2. K. Sun *et al.*, Plasma DNA tissue mapping by genome-wide methylation sequencing for noninvasive prenatal, cancer, and transplantation assessments. *Proc Natl Acad Sci U S A* **112**, E5503-5512 (2015).
3. T. J. Jensen *et al.*, Whole genome bisulfite sequencing of cell-free DNA and its cellular contributors uncovers placenta hypomethylated domains. *Genome Biol* **16**, 78 (2015).
4. X. Ou *et al.*, Epigenome-wide DNA methylation assay reveals placental epigenetic markers for noninvasive fetal single-nucleotide polymorphism genotyping in maternal plasma. *Transfusion* **54**, 2523-2533 (2014).
5. C. Chen *et al.*, A pregnancy with discordant fetal and placental chromosome 18 aneuploidies revealed by invasive and noninvasive prenatal diagnosis. *Reprod Biomed Online* **29**, 136-139 (2014).
6. M. Alberry *et al.*, Free fetal DNA in maternal plasma in anembryonic pregnancies: confirmation that the origin is the trophoblast. *Prenat Diagn* **27**, 415-418 (2007).

7. H. Masuzaki *et al.*, Detection of cell free placental DNA in maternal plasma: direct evidence from three cases of confined placental mosaicism. *J Med Genet* **41**, 289-292 (2004).
8. D. I. Schroeder *et al.*, Early Developmental and Evolutionary Origins of Gene Body DNA Methylation Patterns in Mammalian Placentas. *PLoS Genet* **11**, e1005442 (2015).
9. C. E. Mordaunt *et al.*, Cord blood DNA methylome in newborns later diagnosed with autism spectrum disorder reflects early dysregulation of neurodevelopmental and X-linked genes. *Genome Med* **12**, 88 (2020).
10. C. E. Mordaunt *et al.*, A meta-analysis of two high-risk prospective cohort studies reveals autism-specific transcriptional changes to chromatin, autoimmune, and environmental response genes in umbilical cord blood. *Mol Autism* **10**, 36 (2019).
11. A. Vogel Ciernia *et al.*, Experience-dependent neuroplasticity of the developing hypothalamus: integrative epigenomic approaches. *Epigenetics* **13**, 318-330 (2018).
12. Y. Hu *et al.*, Simultaneous profiling of transcriptome and DNA methylome from a single cell. *Genome Biol* **17**, 88 (2016).
13. M. Hennebelle *et al.*, Linoleic acid-derived metabolites constitute the majority of oxylipins in the rat pup brain and stimulate axonal growth in primary rat cortical neuron-glia co-cultures in a sex-dependent manner. *J Neurochem* **152**, 195-207 (2020).
14. Z. Zhang, C. E. Richardson, M. Hennebelle, A. Y. Taha, Validation of a One-Step Method for Extracting Fatty Acids from Salmon, Chicken and Beef Samples. *J Food Sci* **82**, 2291-2297 (2017).
15. K. Ichihara, Y. Fukubayashi, Preparation of fatty acid methyl esters for gas-liquid chromatography. *J Lipid Res* **51**, 635-640 (2010).
16. E. D. Dodds, M. R. McCoy, L. D. Rea, J. M. Kennish, Gas chromatographic quantification of fatty acid methyl esters: flame ionization detection vs. electron impact mass spectrometry. *Lipids* **40**, 419-428 (2005).
17. Y. Hasegawa *et al.*, Optimization of a Method for the Simultaneous Extraction of Polar and Non-Polar Oxylipin Metabolites, DNA, RNA, Small RNA, and Protein from a Single Small Tissue Sample. *Methods Protoc* **3** (2020).
18. J. Sotelo-Orozco, S. Y. Chen, I. Hertz-Picciotto, C. M. Slupsky, A Comparison of Serum and Plasma Blood Collection Tubes for the Integration of Epidemiological and Metabolomics Data. *Front Mol Biosci* **8**, 682134 (2021).
19. J. T. Smilowitz *et al.*, The human milk metabolome reveals diverse oligosaccharide profiles. *J Nutr* **143**, 1709-1718 (2013).
20. C. M. Slupsky *et al.*, Investigations of the effects of gender, diurnal variation, and age in human urinary metabolomic profiles. *Anal Chem* **79**, 6995-7004 (2007).

Reviewers' comments:

Reviewer #1 (Remarks to the Author):

The authors of the article have addressed my concerns, to the extent that they can.

Reviewer #2 (Remarks to the Author):

The authors have been responsive to my critiques. I have no further suggestions.

Reviewer #3 (Remarks to the Author):

The authors have satisfactorily answered my comments in their revision.

Reviewer #4 (Remarks to the Author):

No further comments.